# Selective attenuation of Ether-a-go-go related K⁺ currents by endogenous acetylcholine reduces spike-frequency adaptation and network correlation

Edward D Cui, Ben W Strowbridge*

Department of Neurosciences, Case Western Reserve University, Cleveland, United States

**Abstract** Most neurons do not simply convert inputs into firing rates. Instead, moment-to-moment firing rates reflect interactions between synaptic inputs and intrinsic currents. Few studies investigated how intrinsic currents function together to modulate output discharges and which of the currents attenuated by synthetic cholinergic ligands are actually modulated by endogenous acetylcholine (ACh). In this study we optogenetically stimulated cholinergic fibers in rat neocortex and find that ACh enhances excitability by reducing Ether-à-go-go Related Gene (ERG) K⁺ current. We find ERG mediates the late phase of spike-frequency adaptation in pyramidal cells and is recruited later than both SK and M currents. Attenuation of ERG during coincident depolarization and ACh release leads to reduced late phase spike-frequency adaptation and persistent firing. In neuronal ensembles, attenuating ERG enhanced signal-to-noise ratios and reduced signal correlation, suggesting that these two hallmarks of cholinergic function in vivo may result from modulation of intrinsic properties.

DOI: https://doi.org/10.7554/eLife.44954.001

*For correspondence:
bens@case.edu

**Competing interests:** The authors declare that no competing interests exist.

## Introduction

Understanding how modulatory systems function to govern neural circuits is a fundamental question in neuroscience. Abundant work (*Sidiropoulou et al., 2009*, *Zhang et al., 2013*, *Dembrow and Johnston, 2014*) has demonstrated that many neuromodulators, such as acetylcholine (ACh), increase the excitability of a wide variety of neurons under both in vitro (*Krnjević et al., 1971*, *Egorov et al., 2002*, *Egorov et al., 2006*, *Rahman and Berger, 2011*, *Knauer et al., 2013*) and in vivo conditions (*Metherate et al., 1992*, *Metherate and Ashe, 1993*, *Goard and Dan, 2009*, *Zhou et al., 2011*, *Pinto et al., 2013*). Yet, studies of circuit activity during tasks that promote ACh release demonstrate more complex changes than expected from a uniform increase in neuronal excitability, including network desynchronization (*Metherate et al., 1992*, *Kalmbach et al., 2012*, *Pinto et al., 2013*, increasing signal-to-noise ratios (SNR, *Zinke et al., 2006*, *Sato et al., 1987*, *Minces et al., 2017*), and reducing correlation of activity levels between neurons (*Goard and Dan, 2009*, *Minces et al., 2017*). Together, these effects enable cholinergic modulation to improve visual discrimination (*Pinto et al., 2013*), enhance sensory coding (*Goard and Dan, 2009*) and influence attention (*Steinmetz et al., 2000*, *Briggs et al., 2013*) and working memory (*Goard and Dan, 2009*, *Ballinger et al., 2016*).

The specific cellular mechanisms that enable ACh to have such wide-ranging and important functional effects remain unresolved despite extensive efforts to define targets of mAChRs modulation. Previous work has identified large set of outward and inward currents that could explain the increase in excitability traditionally found with cholinergic stimulation (*McCormick and Prince, 1985*, *Haj-*

*Dahmane and Andrade, 1996*, *Buchanan et al., 2010*, *Cui and Strowbridge, 2018*). However, two critical issues have prevented unambiguously assigning specific in vivo cellular and circuit-level changes to a specific biophysical mechanism, such as attenuation of the M current. First, studies employing prolonged application of exogenous agonists can trigger different intracellular signaling cascades than the endogenous ligand does when released synaptically (*Davis et al., 2010*, *Wacker et al., 2017*). This limitation suggests that the range of potential mechanisms defined by studies of bath application of carbachol (CCh) and related agonists likely includes many non-physiological targets. Second, the specific modulatory pathways engaged are likely governed by the timing relationship between ACh release and neuronal/synaptic stimulation. Stimulation-evoked discharges in neocortical pyramidal cells typically have strong spike frequency adaptation (SFA) and, to our knowledge, no group has assayed the effects of endogenous ACh on SFA. Previous modeling results, however, suggest that modulation of SFA alone is sufficient to account for both the increased excitability observed at the single cell level as well as many of the network changes (increased SNR, decreased signal correlation, *Wang et al., 2014a*). Acetylcholine also influences synaptic function that could contribute to SNR enhancement and reduced network correlation (*Min et al., 2018*, *Litwin-Kumar et al., 2011*, *Picciotto et al., 2012*). However, these synaptic changes must occur within the context of altered intrinsic physiology, including adaptation dynamics, necessitating an integrative approach that includes determining the primary biophysical targets of cholinergic modulation (*Marder et al., 1996*).

In the present study, we determined the sequence of activation of three primary $K^+$ currents that appear to govern SFA in neocortical pyramidal cells (SK, M, and ERG in their order of activation). Endogenous ACh released by optogenetic stimulation modulates neuronal discharges by selectively attenuating one of these currents (ERG). Acetylcholine acting through mAChRs enhanced intrinsic excitability by selectively reducing the late phase of SFA normally mediated by ERG. The ACh-triggered changes in excitability and SFA were occluded by three different ERG channel antagonists. Examining consequences of ERG modulation by ACh in virtual ensembles, we find an enhancement in SNR and reduction in signal correlation among neurons similar to that observed in vivo following cholinergic stimulation. These results suggest that changes in population neuronal activity associated with common behavioral phenomenon such as attention (*Steinmetz et al., 2000*, *Briggs et al., 2013*) and working memory (*Goard and Dan, 2009*, *Ballinger et al., 2016*) could reflect changes of intrinsic biophysical properties of individual cells that are modulated by cholinergic input. Given the established role of ERG mutations in schizophrenia (*Huffaker et al., 2009*) along with the sensitivity of ERG to many common neuroleptic drugs (*Shepard et al., 2007*), this biophysical SFA modulation may help explain the association between cholinergic dysfunction and schizophrenia (*Sarter and Bruno, 1998*).

## Results

### Increased excitability triggered by endogenous acetylcholine

While many studies (*Krnjević et al., 1971*, *McCormick and Prince, 1986*, *Andrade, 1991*, *Rahman and Berger, 2011*) have examined the effects of continuously applied cholinergic receptor agonists, less is known at a cellular level about the changes in neocortical function following precisely-timed release of endogenous ACh. We assayed the effects of endogenous ACh release in 154 L5 regular spiking neocortical pyramidal cells in P35-45 rats injected with AAV-ChR2 in the nucleus basalis (NB), a major source of cholinergic input to the neocortex in both rodents (*Mesulam et al., 1983a*) and primates (*Mesulam et al., 1983b*). Combining light-stimulated ACh release (20 Hz trains of 2 ms light pulses) with depolarizing steps consistently increased the number of APs evoked by depolarizing stimuli (observed in 84% of experiments). As demonstrated in the example recording shown *Figure 1A–B*, the increase in excitability evoked by endogenous ACh resulted in 39% more spikes within the same 2 s depolarizing step (N = 5 cells; quantified in *Figure 1C–D*). The ACh-mediated excitability increase was abolished by the muscarinic receptor antagonists atropine (10 $\mu$M; N = 5; *Figure 1C–D*, right panels) and pirenzepine (10 $\mu$M; N = 4) demonstrating that neurotransmitters besides ACh that were potentially co-released by ChR2 stimulation of NB axons (*Rye et al., 1984*) were not required to enhance excitability.

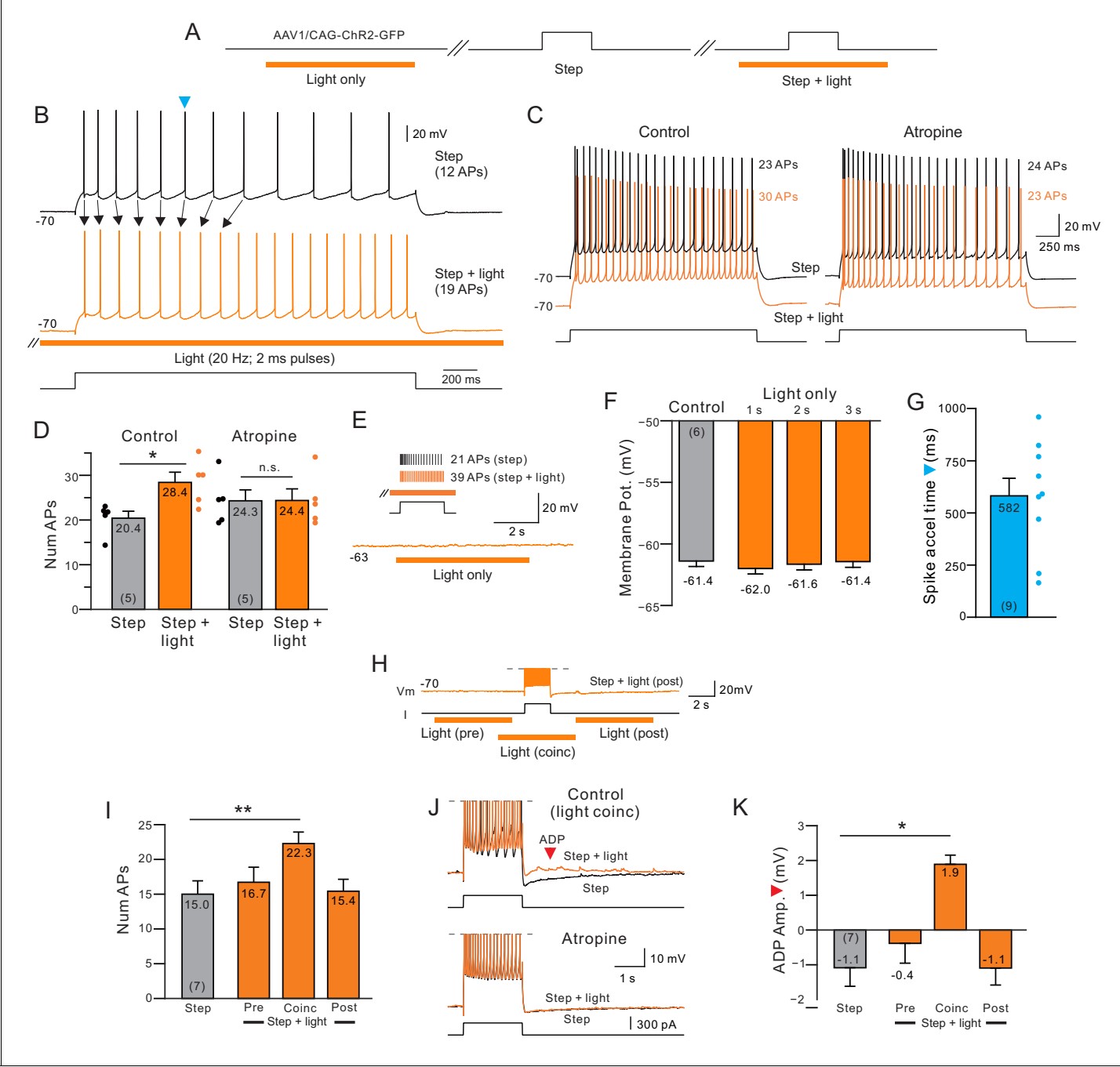

**Figure 1.** Delayed increase in intrinsic excitability following muscarinic receptor activation by endogenous acetylcholine. (**A**) Diagram of experiment protocol; orange bar indicates timing of light pulses that trigger ACh release via ChR2. (**B**) Example step responses in control conditions (black trace) and when ACh was released 2 s before and during the step response (orange trace). Blue arrowhead indicates timing of first AP in the control step response that was accelerated when ACh was released (Quantified in G). (**C**) Blockade of muscarinic cholinergic receptors with 10 $\mu$M atropine abolished the increase in excitability triggered by ACh release. Black traces acquired in control conditions; orange traces illustrate responses with coincident light stimulation to the same depolarizing step. (**D**) Summary of the average number of APs evoked by depolarizing steps in five experiments similar to C. *p=0.0106, T = 4.526; n.s., p=0.847, T = 0.206. paired t-test. (**E**) Example trace showing that ACh release in the absence of a depolarizing step does not modulate the membrane potential. Inset, in the same cell, pairing ACh release with depolarizing step increases the number of APs triggered. (**F**) Comparison of mean membrane potential in control conditions (black bar) and after different durations of light pulse trains only (orange bars; p>0.05; paired t-test). Experiments conducted on neurons in which coincident light trains increased the number of APs triggered by depolarizing steps. (**G**) Summary plot of the average time during similar step responses before an AP was accelerated (shorter latency for the Nth AP when ACh was released, compare B for example traces). (**H**) Example illustrating different light/step timing protocols. Response shown (orange trace) is from 'post'
*Figure 1 continued on next page*

*Figure 1 continued*

protocol. (I) Coincident ACh release is required to increase neuronal excitability. Plot of the number of APs evoked the same depolarizing step in control conditions (black bar) and when a 6 s light pulse train was applied 7 s before (Pre), 2 s before (Coinc) or 2 s after (Post) the depolarizing step. \*\*p=0.00334, T = 5.8373; paired t-test, Bonferroni corrected. (J) Coincident ACh release and depolarizing step stimuli reveals an afterdepolarization potential (ADP; red arrowhead) that is abolished by 10 $\mu$M atropine. Action potentials truncated. (K) Summary plot of the dependence of the ADP on coincident ACh release. Same light train/depolarizing step timing as G. \*p=0.0219, T = 3.977; paired t-test, Bonferroni corrected. Plots are mean ± SEM unless noted throughout study.

DOI: https://doi.org/10.7554/eLife.44954.002

The following figure supplements are available for figure 1:

**Figure supplement 1.** Control experiments for ChR2 stimulation.

DOI: https://doi.org/10.7554/eLife.44954.004

**Figure supplement 2.** Minimal effect of changing light pulse frequency used to stimulate ChR2.

DOI: https://doi.org/10.7554/eLife.44954.005

**Figure supplement 3.** Calibration of stereotaxic coordinates used for AAV injections in NB.

DOI: https://doi.org/10.7554/eLife.44954.003

ChR2 stimulation of NB axons alone (without a temporally overlapping intracellular depolarization) did not modulate the membrane potential over the 2 s window we typically used to assay excitability changes or directly trigger discharges (*Figure 1E–F* and *Figure 1—figure supplement 1A*). We found no statistically significant changes in neuronal input resistance following light pulse trains presented alone (*Figure 1—figure supplement 1B*). In addition, we found no effect of light pulse trains on depolarizing step responses in four neocortical pyramidal cells from age-matched control animals without AAV-ChR2 injections (no significant difference in membrane potential modulation or the number of spikes evoked by depolarizing steps, both p>0.05, paired t-test).

The increase in neuronal excitability we find with paired ACh release was repeatable across multiple trials with no modulation of excitability observed in interleaved control trials without ChR2 stimulation (*Figure 1—figure supplement 1C*). The excitability increase also appeared to be a direct action on the recorded neuron rather than from an enhancement of synaptic excitation since the excitability modulation was unaffected by bath application of antagonists for ionotropic glutamate and GABA receptors (*Figure 1—figure supplement 1D*; N = 3).

Unexpectedly, the increase in neuronal excitability elicited by ACh was not uniform during the response to the depolarizing step. Instead, release of endogenous ACh selectively advanced the timing of APs triggered after ~ 500 ms from the step onset (*Figure 1B and G*). The increase in step response excitability combined with minimal changes in resting conductance suggests that a main action of ACh is to modulate cellular processes activated by depolarizing stimuli. The primary goal of the subsequent experiments in this study is to define the cellular basis for the delayed increase in excitability in response to simple depolarizing step stimuli. We then use correlation analysis of single cell responses and computer simulations to determine the likely effects of ACh on neural ensembles.

## Mechanism of cholinergic modulation of neuronal excitability

In these initial experiments, ACh was released by light trains that began 2 s before the depolarizing step and ended 2 s following step offset. In seven experiments, we systemically varied the timing of the light pulses relative to the depolarizing step (*Figure 1H–K*). The only timing paradigm that increased the number of spikes evoked occurred when current steps temporally overlapped the light stimuli. ChR2 stimulation before the current had no direct effect on the step response and ChR2 stimulation following the step failed to trigger discharges (*Figure 1I*). Throughout this study, we use the term 'coincident ACh release' to represent current steps during periods of light stimulation. In some coincident experiments, we varied the onset of the light stimulus ACh but we always ensured that light pulses were applied throughout the duration of the step. Only ACh release that was coincident with the current step modulated the afterpotential, converting the afterhyperpolarization into an afterdepolarization (*Figure 1J–K*). Like the enhancement in excitability, the afterdepolarization following coincident step and ChR2 stimulation was abolished by atropine (*Figure 1—figure supplement 1E*). The magnitude of light-evoked enhancement of neuronal excitability (*Figure 1—figure supplement 2A–B*) and afterpotential modulation (*Figure 1—figure supplement 2C*) was unaffected when steps were paired with light pulse trains whose frequency varied from 5 to 40 Hz.

The delayed increase in excitability we find with ChR2 stimulation appeared to reflect both an increase in average spike frequency (shown in *Figure 1D and I*) and a change in spike frequency adaptation (SFA). Instead of assaying SFA at only two time points (by comparing early and late ISIs within the step response; *Ji et al., 2009*, *Vandecasteele et al., 2011*), we assayed SFA continuously by dividing the 2 s step response into 40 bins (each 50 ms in duration) and plotting the cumulative number of spikes triggered in each successive bin. This method both reveals the dynamic pattern of SFA throughout the response and enables summary plots containing multiple trials to be averaged. The SFA expected for regular spiking neurons (*Connors and Gutnick, 1990*, *Dégenètais et al., 2002*) is indicated in these plots by sub-linear cumulative spike count functions (black plot in *Figure 2A*).

Releasing ACh via light stimulation of ChR2 altered SFA kinetics starting approximately 800 ms into the step response (*Figure 2B*, left). In this and subsequent plots, we quantified the onset of SFA modulation by determining the first time bin where the cumulative spike count was increased by at least 1 AP. This $\Delta$1AP time, indicated by orange arrows in *Figure 2B*, was near the obvious bifurcation between the step only (black) and step + light (orange) cumulative spike count plots. The onset of ACh-mediated SFA modulation appeared to be independent of discharge frequency when responses to different current step amplitudes were compared in the same experiment (*Figure 2B*; $\Delta$1AP times not different; p=0.664, T = 0.453; paired t-test). However, the degree of excitability enhancement did vary between weak and strong depolarizing steps with more additional spikes produced when ChR2 stimulation was combined with strong steps than with weak steps (*Figure 2B*, inset). We found similarly delayed $\Delta$1AP times (>800 ms) when we varied the frequency of light pulses that triggered ACh release between 5–40 Hz. There were no statistically significant differences in the increase in number of spikes evoked by steps paired with light trains of different frequencies (*Figure 1—figure supplement 2B*).

The delayed increase in excitability in response to moderate duration (2 s) steps suggests that release of endogenous ACh with ChR2 should selectively modulate responses to long duration steps while leaving responses to short duration stimuli unaffected. We confirmed that prediction by testing responses to 100 ms step stimuli and found no increase in the number of APs evoked with stimulation (6.0 ± 0.0 with step only vs 6.0 ± 0.0 with step + light; N = 5). In the same 5 cells, pairing light stimulation with longer duration steps (750–2000 ms) increased the number of APs directly demonstrating the specificity of ChR2 modulation to long-duration responses in these experiments (20.2 ± 0.7 with step only vs. 24.8 ± 0.4 with step + light; N = 5; p<0.001; paired t-test).

Two different mechanisms can explain the pronounced delay before ChR2 stimulation increased the spiking rate. This delay could reflect either the time required to initiate muscarinic receptor signaling pathways (*Figure 2C*, red arrow) or it could reflect fast signaling but slow kinetics of underlying ionic current responsible for modulating firing rates (*Figure 2C*, blue arrow). To discriminate between these possibilities, we varied the latency between the onset of ChR2 light stimulation and the depolarizing step. If the delay before individual spike times advanced reflects simply the time required for muscarinic receptor to activate a signaling cascade, then the bifurcation time should change when ChR2 was activated earlier. By contrast, if the rate limiting process is the modulated ion channel, then the bifurcation time should be insensitive to when ACh release was initiated as long as it occurred far enough before the step to engage the signaling cascade.

We found that modulating the delay between ChR2 stimulation and the depolarizing step by 3 s had no significant effect on either the enhancement in neuronal excitability or SFA dynamics. As shown in *Figure 2D*, increasing or decreasing the ChR2-to-step latency by 1 s from the standard 2 s latency tested above varied the average $\Delta$1AP time by < 100 ms (lack of effect on excitability summarized in *Figure 2E* and $\Delta$1AP times in *Figure 2F*).

While experiments to determine the signaling pathways responsible for muscarinic modulation of excitability are beyond the scope of the present study, we conducted a separate set of experiments in which light pulses were applied mid-way through the depolarizing step response to estimate the latency of the intracellular signaling cascade. We used a longer depolarizing step in these experiments (4 s duration) since our results thus far suggest endogenous ACh modulates a current with slow kinetics; we wanted to avoid activating mAChRs while the target current was activating rapidly early in the step response. As shown in the example trace in *Figure 2—figure supplement 1A*, stimulating ChR2 mid-way through the step depolarization still increased the number of APs evoked (quantified in *Figure 2—figure supplement 1B* ). We estimated the latency from the initial ChR2

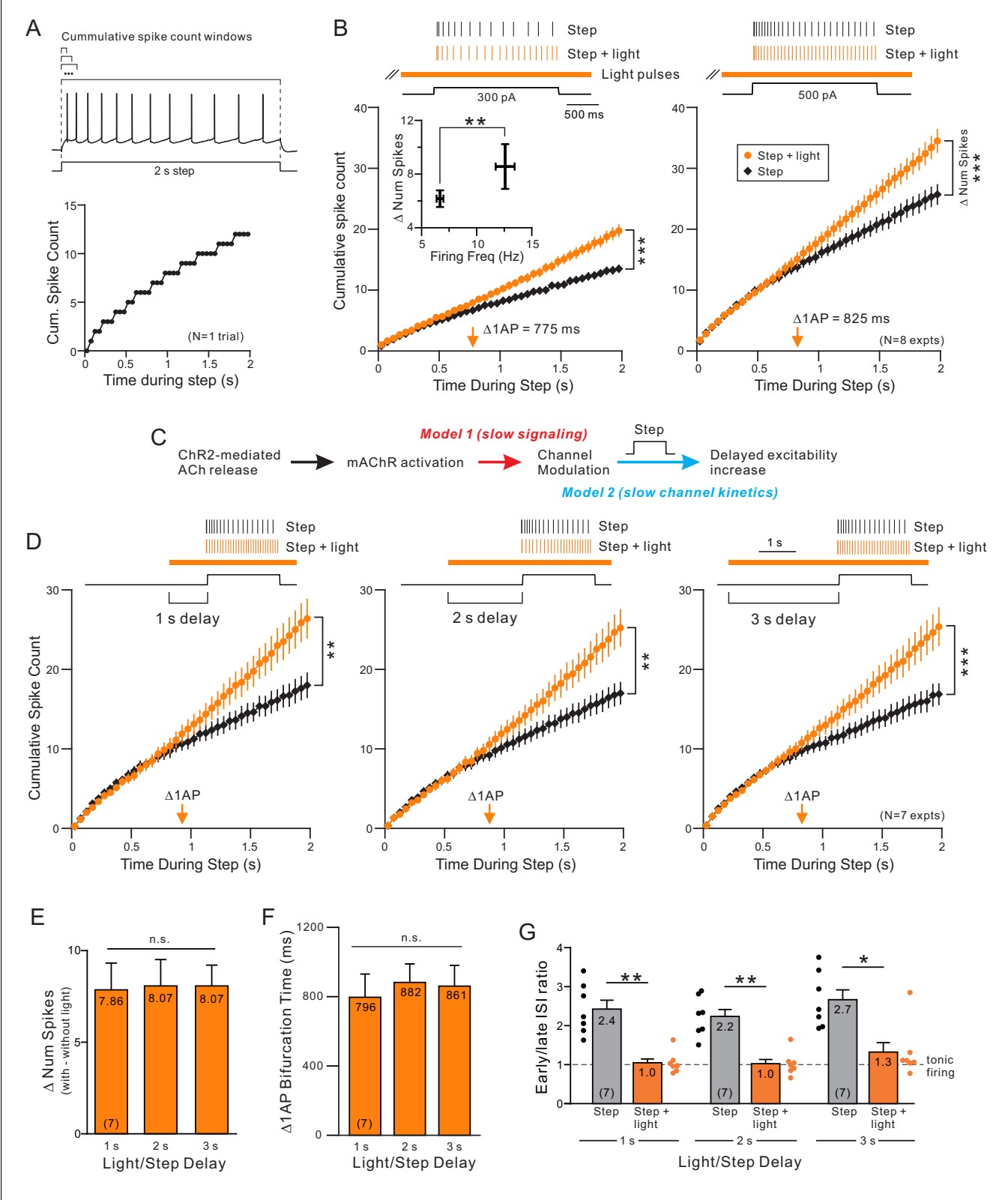

**Figure 2.** Endogenous ACh enhances excitability by attenuating a slow leak K$^+$ current. (**A**) Example step response illustrating increasing duration analysis windows (above trace) used to compute the cumulative spike count for that one response (bottom). (**B**) Coincident ACh increases neuronal excitability in response to different step amplitudes. Plot of the cumulative spike count across different durations of analysis windows for weak (left panel) and moderate (right panel) step amplitudes. Individual points represent mean ± SEM over eight experiments with typically three trials per

*Figure 2 continued on next page*

*Figure 2 continued*

experiment in control (black) and ACh release (orange, light pulse train initiated 2 s before step). Example spike discharges evoked by the same depolarizing step shown above plot with vertical lines indicating the timing of each AP relative to the step. The total number of APs evoked by the step (right-most points in summary plot) differed significantly in response to both weak and moderate step intensities; *** 300 pA steps (left); p=6.04E-5, T = 8.530; 500 pA steps (right): p=7.31E-6, T = 11.754; both paired t-test. Inset: The increase in number of spikes between step and step + light vs. firing frequency during step only. *p=0.0130, T = 3.304, paired t-test. Orange arrow near X axis indicates time within depolarizing step responses when ACh increased the average cumulative spike count by 1 AP. The same analysis format was used in B, D, and in subsequent figures. (C) Diagram illustrating two potential mechanisms to explain long delay before ACh enhances neuronal excitability. The critical signaling event triggered by muscarinic receptors could be slow (red text), resulting in the ~ 700–900 ms delay, or the delay could reflect slow intrinsic kinetics of the ionic channels modulated by ACh (blue text). (D) ACh-modulated increase in neuronal excitability does not reflect muscarinic receptor signaling time. Plot of cumulative spike count in control conditions and with coincident ACh release (as in B) in seven experiments in which the light pulse train began 1 s (left panel), 2 s (middle) or 3 s (right) before the depolarizing step. **p<0.01 in all three panels, paired t-test; 1 s delay: T = 4.60.5973, 2 s: T = 5.199, 3 s: T = 6.579. E-G, Summary of the seven experiments illustrated in D. (E) The increase in number of spikes from the step to step + light conditions (illustrated as Δ number of spikes) was not significantly different between different delays (n.s. p>0.05, 1 s vs. 2 s: p=1, T = 0.235, 1 s vs. 3 s: p=1, T = 0.328, 2 s vs. 3 s: p=1, T = 0. Paired t-test, Bonferroni corrected). (F) Plot represents mean bifurcation time (time when the step and step + light cumulative spike count plots diverged by at least 1 AP) calculated separately in each experiment and then averaged. (Bifurcation times presented in B and D represent times when the ensemble average plots diverged by at least 1 AP. Both bifurcation time protocols typically yielded similar latency estimates.) (G) Mean ISI ratio that assesses spike frequency adaptation across conditions. ISI ratio was calculated by the ratio between the average of the last 2 ISIs and the average of the first 2 ISIs. ISIs were significantly reduced by light induced ACh release (step vs. step + light: ** 1 s: p=0.00140, T = 7.808; ** 2 s: p=0.00206, T = 7.277; *p=0.0167, T = 4.874; paired t-test, Bonferroni corrected). Within step + light conditions, all comparisons between delays were not significantly different (1 s vs. 2 s: p=1, T = 0.520; 1 s vs. 3 s: p=0.936, T = 1.620; 2 s vs. 3 s: p=0.798, T = 1.735; paired t-test, Bonferroni corrected).

DOI: https://doi.org/10.7554/eLife.44954.006

The following figure supplement is available for figure 2:

**Figure supplement 1.** Estimation of muscarinic receptor signaling latency to attenuate ERG.

DOI: https://doi.org/10.7554/eLife.44954.007

light stimulus to the functional effect of the signaling cascade (increased firing rate) in two ways. First, we determined the first inter-spike interval (ISI) that was reduced following ChR2 stimulation (compared with the mean ISI immediately before light stimulation). This ISI-based analysis method likely provides a lower bound on the signaling latency (~520 ms; *Figure 2—figure supplement 1C*). We also estimated an upper bound on the signaling latency by computing the Δ1AP time, paralleling the analysis in *Figure 2C*. This latency, 675 ms (*Figure 2—figure supplement 1D*) is only modestly longer than the lower bound estimate and suggest that ChR2-stimulated ACh release can modulate channel function and increase neuronal excitability within 500–700 ms.

Finally, we confirmed using conventional early/late ISI ratio measurements that under all three timing conditions ChR2 reduced SFA and promoted near tonic firing (*Figure 2G*). These results suggest that the delayed effect of ChR2-stimulated ACh release likely reflects slow kinetics of the ionic currents modulated by ACh.

## Endogenous ACh modulates ERG channels

The relatively constant Δ1AP time metric we find in *Figure 2D–F* suggests that the delay before the spiking rate increases with coincident ACh release reflects the slow kinetics of a modulated ion channel active near rest (and thus contributing to SFA). Ether-à-go-go Related Gene (ERG) K$^+$ channel is likely a potential candidate as we (*Cui and Strowbridge, 2018*) and others (*Saganich et al., 2001*, *Papa et al., 2003*) previously demonstrated that ERG is expressed in L5 neocortical pyramidal cells and functions to dampen neuronal excitability. We also recently demonstrated (*Cui and Strowbridge, 2018*) that activation of mAChRs with bath CCh can inhibit ERG. When tested using voltage-clamp steps from the resting potential of L5 neurons, whole-cell ERG-mediated currents take hundreds of ms to develop (*Cui and Strowbridge, 2018*, *Niculescu et al., 2013*), mimicking the slow SFA kinetics we find.

If cholinergic attenuation of ERG underlies the reduction in SFA following ChR2 stimulation, then pharmacological blockade of ERG channels should occlude the effects of endogenous ACh release since the modulated channel is already blocked. As shown in *Figure 3A–B*, we find that blocking ERG with terfenadine (Terf) increased baseline excitability during control conditions (without ACh release) and did indeed occlude the light-stimulated enhancement of the spike discharge (no

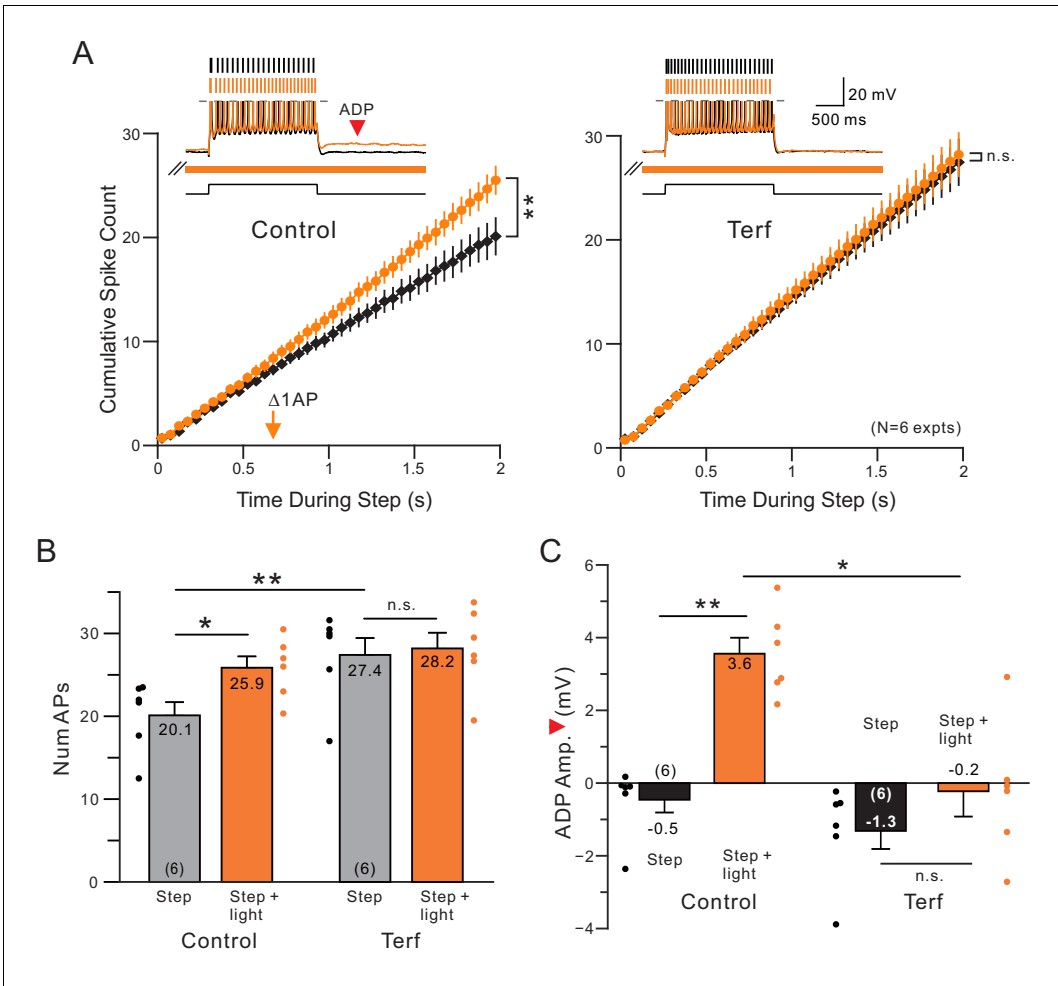

**Figure 3.** Terfanadine (10 $\mu$M) abolished the increase in neuronal excitability normally observed following coincident ACh release. (**A**) Control and Terf responses from the same neurons with identical light pulse trains timing in each condition. \*\*p=0.00201, T = 5.884; n.s. p=0.254, T = 2.063. (**B**) Quantification of total number of APs across conditions. Total number of APs was increased upon ACh release (\*p=0.0158, T = 4.719). This effect was blocked by application of terfenadine (n.s. p=1, T = 0.952). Application of Terfenadine significantly increased the total number of APs without light (step, Control vs. Terf: \*\*p=0.00498, T = 6.141; all paired t-test, Bonferroni corrected). (**C**) plot of reduction in afterdepolarization response amplitude (indicated by red arrowhead in example Control step + light orange trace) by Terf. \*\* Control step vs step + light: p=0.00992, T = 5.257; n.s. Terf step vs step + light P=0.267, T = 2.107; \* Control step + light vs Terf step + light: p=0.0142, T = 4.843; all paired t-tests, Bonferroni corrected.

DOI: https://doi.org/10.7554/eLife.44954.008

statistically significant increase in number of APs when light and the step were presented together). Blockade of ERG also reduced the ADP response evident following the step response when ACh was present (orange arrowhead in example trace in *Figure 3A*; ADP quantified in *Figure 3C*). Two other chemically different ERG antagonists, E-4031 and ErgToxin, also abolished the ACh-mediated enhancement in step responses (N = 3 for each antagonist; light stimulation evoked 5–6 additional spikes in the step response in each experiment before blocking ERG and on average < 1 additional spike after ERG blockade). By contrast, antagonists of both the M current and SK channels failed to occlude the light-stimulated enhancement of step discharges (M current:~5 extra spikes evoked by step + light both with and without XE991, N = 3; SK: 4–7 extra spikes evoked both with and without NS8593, N = 3). These results suggest that endogenous ACh released by light activation of ChR2 expressed in cholinergic neuron axons may act specifically to attenuate ERG currents, enhancing firing during step responses and accounting for the delayed reduction in SFA in *Figures 1–2*.

We next asked if ChR2-stimulated ACh release modulated voltage-clamped K$^+$ currents in neocortical pyramidal cells. We employed a similar protocol used in our prior voltage-clamp analysis of ERG modulation using a cholinergic agonist (*Cui and Strowbridge, 2018*) except that intracellular QX-314 was substituted for bath TTX to block voltage-gated Na$^+$ channels. We were not able to block voltage-gated Na$^+$ or Ca$^{2+}$ channels using bath perfusion in the present study since these currents facilitate ChR2-driven ACh release.

Over ten similar experiments, ChR2-driven ACh release selectively reduced the late phase of the outward currents evoked by depolarizing steps (1802 ± 312 pA in control to 1522 ± 293 pA with ChR2 stimulation; p=3.23e-05, T = 7.630) without affecting the initial current response assayed over the first 100 ms (1354 ±381 pA in control vs 1326 ± 380 pA; p>0.05; T = 1.772; both paired t-tests; *Figure 4A*). The late-developing ACh-modulated outward current with slow rising phase kinetics was revealed by subtracting interleaved light stimulated ChR2 and step-only trials (*Figure 4B*). On average, the ChR2-sensitive outward current began ~ 0.5 s after the depolarizing step onset (summary plot in *Figure 4C*) and developed with kinetics well fit by single exponential functions (mean tau = 1.2 s; *Figure 4D*), consistent with previous work on ERG kinetics (*Sturm et al., 2005*, *Wang et al., 1996*, *Wang et al., 1997a*). The late onset of the ChR2-sensitive current closely

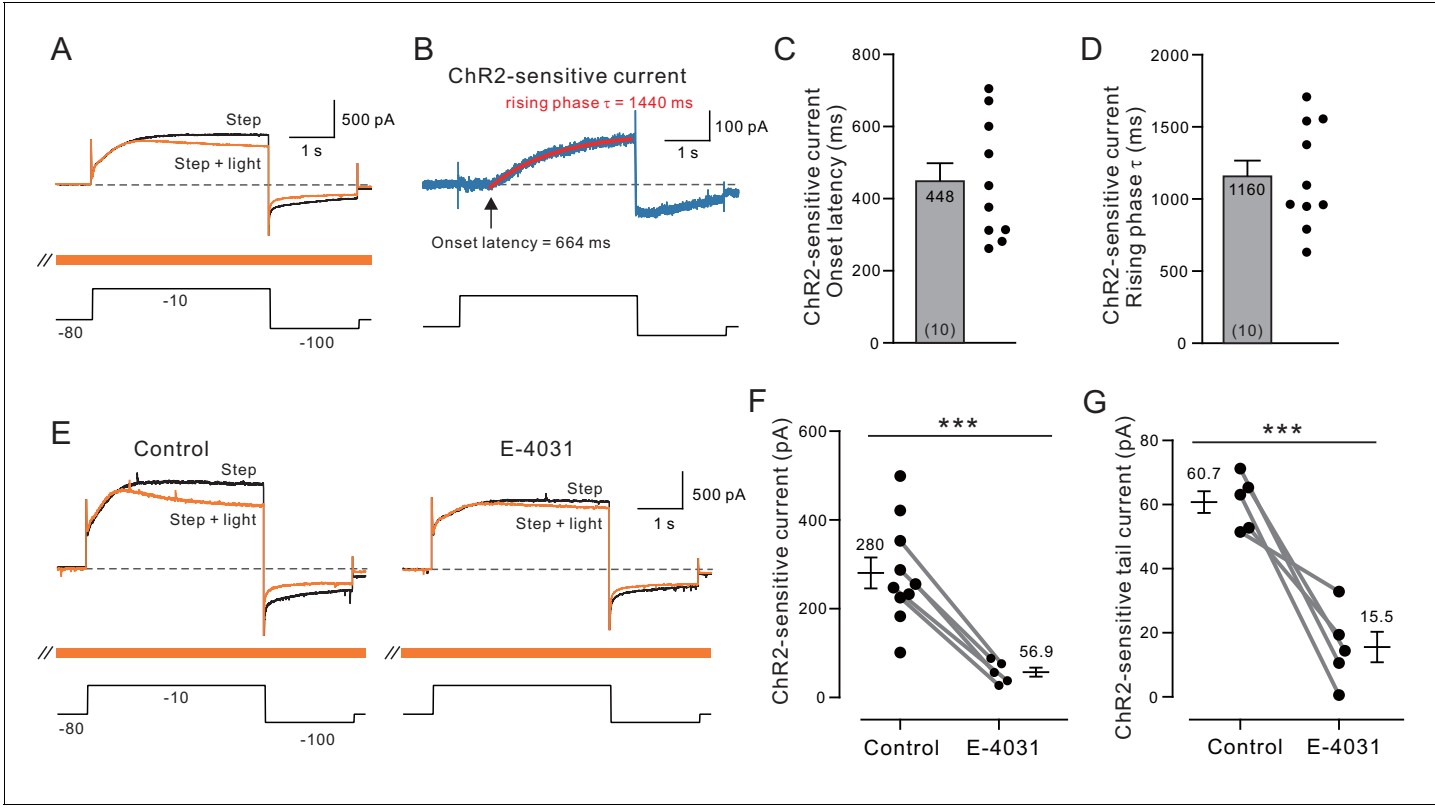

**Figure 4.** Voltage-clamp analysis of ACh-sensitive outward current. (**A**) ChR2 stimulation with light selectively reduces a late-developing outward current. Black trace represents control response to step protocol without ChR2 stimulation; orange trace shows response when applied with ChR2 stimulation initiated 1.5 s before step to −10 mV and continued throughout step protocol (orange bar). (**B**) Subtraction of the two traces in A reveals outward current attenuated by ChR2 light stimulation. Single exponential fit superimposed in red; arrow indicates onset latency estimated from 0 crossing of exponential fit function. (**C**) Summary of onset latency of ACh-sensitive outward current in 10 experiments. (**D**) Summary of tau of exponential fits to subtracted current rising phase in the same 10 experiments. (**E**) ERG blocker E-4031 attenuates an overall outward current response and occludes most of the modulation by ChR2 light stimulation (different cell from **A-B**). (**F**) Summary of maximal ChR2-sensitive outward current in 10 experiments calculated as the mean of final 100 ms of step response to −10 mV in control conditions and five experiments that tested E-4031; solid lines indicate subset of control points tested in E-4031. ***p=0.0011, T = 4.175 (unpaired comparison including all Ns); p=0.00014, T = 14.211 (paired comparison of 5 cells tested in both control and E-4031 conditions). (**G**) Plot of reduction in tail component of the ChR2-sensitive current by E-4031. Tail current magnitude estimated by following the step to −100 mV. ***p=0.00581, T = 5.269, paired t-test. Membrane potentials indicated in A and E are corrected for the liquid junction potential. Capacitive transients truncated in current traces.
DOI: https://doi.org/10.7554/eLife.44954.009

matches the delayed onset of ERG-mediated SFA in related current clamp recordings shown in *Figures 1–3*.

Both the ChR2-sensitive outward current evoked by depolarizing steps to −10 mV and the tail current evoked by subsequent hyperpolarizing steps to −100 mV were attenuated by the ERG antagonist E-4031 (10 $\mu$M; *Figure 4E*). *Figure 4F* summarizes the reduction in maximal light-sensitive outward current by E-4031 (to ~ 20% of control magnitude). *Figure 4G* shows a similar reduction in tail current magnitude (to 26% of control magnitude) by E-4031. Together these results support the model indicated by assaying step-evoked spike discharges in which ChR2 stimulation of ACh release attenuates a late-developing outward current with intrinsically slow kinetics. Both the slow rising phase kinetics of the underlying outward current and its sensitivity to the selective ERG blocker E-4031 (*Wang et al., 1997b*) suggest ACh reduces spike frequency adaptation by attenuating ERG current.

## Endogenous ACh selectivity

While our findings showed that ERG antagonists occlude the increase in neuronal excitability triggered by endogenous ACh, previous work using bath application of cholinergic receptor agonists such as CCh have identified other $K^+$ channel targets of mAChR-driven modulation (*Suh and Hille, 2002*, *Buchanan et al., 2010*, *Hirdes et al., 2004*) that likely contribute to SFA in cortical pyramidal cells. Most commonly, previous investigators have focused on SK and M currents (*McCormick, 1992*, *Madison and Nicoll, 1984*, *Storm, 1989*) in context of SFA while a role for ERG modulation in regulating excitability has been demonstrated in neocortical (*Cui and Strowbridge, 2018*) and other neurons (*Chiesa et al., 1997*, *Sacco et al., 2003*, *Pessia et al., 2008*). Presumably all three $K^+$ currents contribute to SFA. However, the differential roles, kinetic properties, and modulation mechanism for these currents have not yet been defined within the same neuron. We next attempted to define the functional signatures of the intrinsic modulation effected by SK, M and ERG currents. Our primary goal is to determine how kinetic properties of these different $K^+$ currents (e.g., when they begin to contribute to SFA) relate to the time scale we find for modulation of SFA by endogenous ACh.

Before conducting tests with pharmacological agents that block different $K^+$ channels, we first used a conductance-based model of pyramidal cells to ask if our cumulative spike count assay was sensitive enough to detect changes in SFA kinetics. This initial computational approach also reveals how changing $K^+$ current dynamics affects experimental measures of SFA such as cumulative spike count assays. In *Figure 5A–B*, we introduced an additional $K^+$ conductance whose kinetics was predefined (e.g., not implemented through H-H type channel kinetics) and which followed either the time course of the current step stimulus ('continuous K') or increased linearly during the step ('Linearly-increasing K'). The maximal conductance of both types of $K^+$ currents was set to reduce the total number of APs evoked during the step by an equivalent amount (~6 APs). Cumulative spike count plots revealed different SFA kinetics generated by each type of $K^+$ conductance with the Δ1AP time when the linearly-increasing $K^+$ current was introduced (925 ms, right panel in *Figure 5B*) versus 275 ms when continuous $K^+$ current was used (left panel). The delayed increase in SFA with the linearly-increasing $K^+$ current was not because of a specific step amplitude used in these simulations. Linearly-increasing $K^+$ currents generated delayed SFA increases across a large range of stimulus amplitudes (all > 600 ms Δ1AP times; *Figure 5—figure supplement 1*). We found the large difference in Δ1AP times with continuous and linearly-increasing $K^+$ conductance persisted when the maximal conductance was varied over a 5-fold range (data not shown).

We next assayed the change in SFA kinetics associated with SK, M and ERG currents using specific pharmacological blockers under standardized conditions (2 s steps that evoked 21 ± 3 APs from −70 mV) before the blockers we applied. Reducing SK currents using NS8593 triggered a rapid increase in excitability with a Δ1AP time of < 100 ms (left panel in *Figure 5C*). Attenuating M currents with XE991 reduced SFA starting ~ 100 ms later, with a Δ1AP time of 225 ms (middle panel). The cumulative spike count profile of the effects of XE991 resembled the addition of the continuous $K^+$ current in the simulations show in *Figure 5A–B*. Finally, reducing ERG current with Terf generated the most delayed change in SFA, with a Δ1AP time of ~ 600 ms (right panel). The normal action of SK currents was reduced by strong intracellular $Ca^{2+}$ buffering in these experiments when assaying changes in SFA with XE991 and Terf. However, we found similar delayed SFA modulation with Terf when SK currents were not attenuated (~800 ms Δ1AP times; stimulus amplitude adjusted to

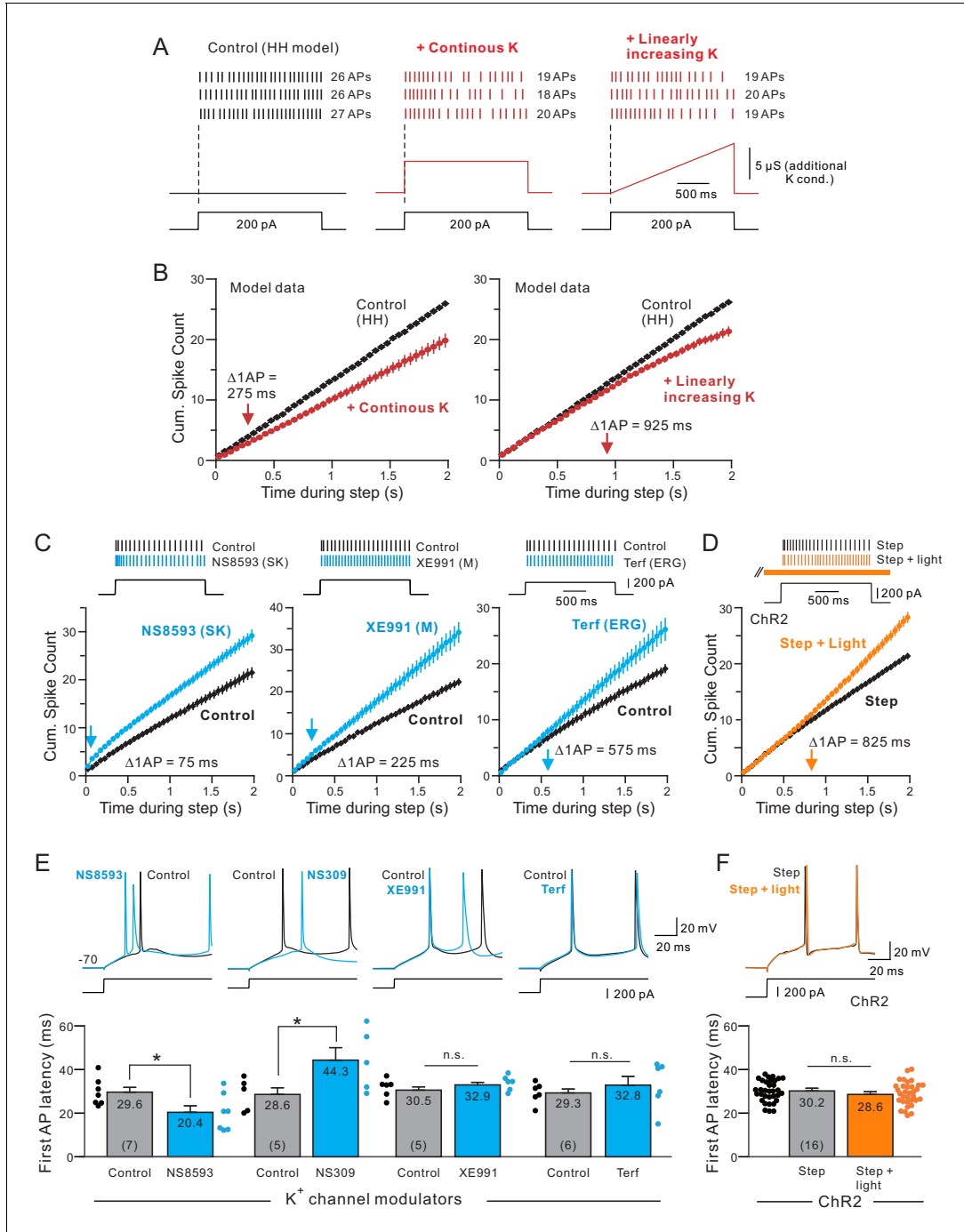

**Figure 5.** Different kinetics of K⁺ currents that mediate spike frequency adaptation in L5 neocortical pyramidal cells. (**A**) Three spike discharge responses from simulations using a Hodgkin-Huxley conductance-based computer model (HH model; black traces in left panel; see Materials and methods for details). Other panels illustrate the change in spike pattern when an additional K⁺ conductance is introduced (red traces, middle: a step increase in K⁺ current that follows the current stimulus timing; right: a linearly increasing K⁺ conductance). (**B**) Including a linearly increasing K⁺ conductance triggers a delayed reduction in neuronal excitability (right panel) that occurs with a longer latency than a step increase in K⁺ conductance (left panel). The magnitudes of the added K⁺ conductances adjusted to generate similar total number of APs in response to a standardized step current stimulus. Bifurcation time indicated by red arrows near X axes indicate when reduction in excitability decreased the cumulative spike count by at least 1 AP. (**C**) Cumulative spike count plots demonstrating the diverse timing of when neuronal excitability increases following blockade of three different K⁺ currents (left panel: SK blocker NS8593, 10 μM; middle: M blocker XE991, 10 μM; right: ERG blocker terfenadine, 10 μM). Bifurcation time, reflecting an increase in the cumulative spike count plot of at least 1 AP (blue arrows), is indicated within each panel. (**D**) Release of endogenous ACh using light pulse trains revealed a bifurcation time (825 ms) longer than any of the K⁺ channel blockers but

*Figure 5 continued on next page*

*Figure 5 continued*

closest to the bifurcation following blockade of ERG (575 ms). Depolarizing step amplitude adjusted to elicit ~ 20 APs in control conditions in each experiment in C-D. (E) Only drugs that blocked SK channels affected the latency to the initial AP in the step response. Bidirectional modulation of first spike latency by the SK blocker (NS8593) and SK activator (NS309, 5 $\mu$M) while blockade of M current with XE991 or ERG with Terf had no effect on first spike latency. * (NS8593) p=0.0130, T = 3.489; * (NS309) p=0.0124, T = 4.324; n.s (XE991) p=0.350, T = 1.030; n.s. (Terf) p=0.364, T = 0.998; all paired t-tests. (F) Release of endogenous ACh had no effect on first spike latency (n.s.: p=0.151, T = 1.514; paired t-test). Experiments assaying bifurcation times and first-spike latency with Terf and XE991 used strong $Ca^{2+}$ buffering (10 mM BAPTA) to attenuate SK-mediated effects. Parallel experiments using weak $Ca^{2+}$ buffering and Rin compensation presented in *Figure 5—figure supplements 1–2* associated with this figure.

DOI: https://doi.org/10.7554/eLife.44954.010

The following figure supplements are available for figure 5:

**Figure supplement 1.** Bifurcation time using constant and linearly-increasing $K^+$ currents in computer simulations.

DOI: https://doi.org/10.7554/eLife.44954.011

**Figure supplement 2.** ERG-blocking experiments with low Ca buffering.

DOI: https://doi.org/10.7554/eLife.44954.012

**Figure supplement 3.** First spike latency was not affected by Terf in experiments with weak $Ca^{2+}$ buffering.

DOI: https://doi.org/10.7554/eLife.44954.013

**Figure supplement 4.** Additional control experiments related to first spike latency.

DOI: https://doi.org/10.7554/eLife.44954.014

compensate for large change input resistance mediated by Terf with low EGTA internal solutions; *Figure 5—figure supplement 2A–B*). These results suggest that SFA in L5 pyramidal cells is normally mediated by the sequential recruitment of SK, M and then ERG $K^+$ currents.

When assayed under conditions as those in *Figure 5C*, endogenous ACh reduced SFA with a long delay (*Figure 5D*; ~800 ms $\Delta 1AP$ time) with cumulative spike count functions that resembled both the pharmacological results with Terf and the simulations with a linearly-increasing $K^+$ conductance (right panel in *Figure 5B* and right panel in *Figure 5C*). When cumulative spike time functions were analyzed separately in each experiment (as in *Figure 2F* when comparing different light/step latencies), we found no statistically significant difference between the mean $\Delta 1AP$ times in ChR2 and Terf experiments (p=0.701, T = 0.392) but highly significant reductions in $\Delta 1AP$ times with XE991 (p=6.205E-4, T = 4.404, Bonferroni corrected) and NS8593 (p=2.145E-4, T = 5.0387, Bonferroni corrected) relative to the mean $\Delta 1AP$ time in ChR2 experiments.

We also determined whether any of the four $K^+$ channel modulators or release of endogenous ACh affected the first AP latency in the same experiments. Only SK modulators affected first spike latency with the SK blocker NS8593 reducing the firing latency while the SK activator NS309 increasing the first spike latency (both p<0.05; *Figure 5E*). Blockade of M current with XE991 and ERG with Terf (*Figure 5E* and *Figure 5—figure supplement 3*) and E-4031 (*Figure 5—figure supplement 4A*) all failed to modulate first spike latency (all p>0.05). We also confirmed that SK modulators but not M or ERG blockers affected subthreshold responses by comparing AHP following brief (30 ms) steps to −50 mV (*Figure 5—figure supplement 4B*). ChR2-stimulated release of endogenous ACh also failed to modulate the first spike time (p>0.05; *Figure 5F*), providing additional evidence that endogenous ACh selectively modulated ERG without affecting SK-mediated currents. The bidirectional modulation of the first spike timing by NS8593 and NS309 implies that SK currents are activated, at least in part, by sub-AP threshold $Ca^{2+}$ influx, perhaps through T-type low-threshold VGCCs. Previous studies (*Wolfart and Roeper, 2002*, *Matschke et al., 2018*) demonstrated close functional coupling between SK channels and T-type channels in other cell types, providing a potential mechanism to explain why SK channel modulators affected first spike latency in our experiments.

Finally, we assayed the SFA dynamics by calculating the slope of the differential cumulative spike count function (illustrated in *Figure 6A*). The motivation for quantifying SFA dynamics during the step response was the apparent reduction in the divergence between control and NS8593 conditions in the plots shown in the left panel of *Figure 5C*. This reduced divergence could reflect the rapid recruitment followed by sustained reduction of a NS8593-sensitive SK current during the step response. As shown in left panel in *Figure 6B*, the rate of SFA reduction assessed using this method was maximal during the first time point assayed (200 ms into the step response) and then decayed until reaching a lower steady-state level at 500 ms into the step response. This presumptive reduction in the SK-mediated adaptation could be explained either by inactivation of the SK channels

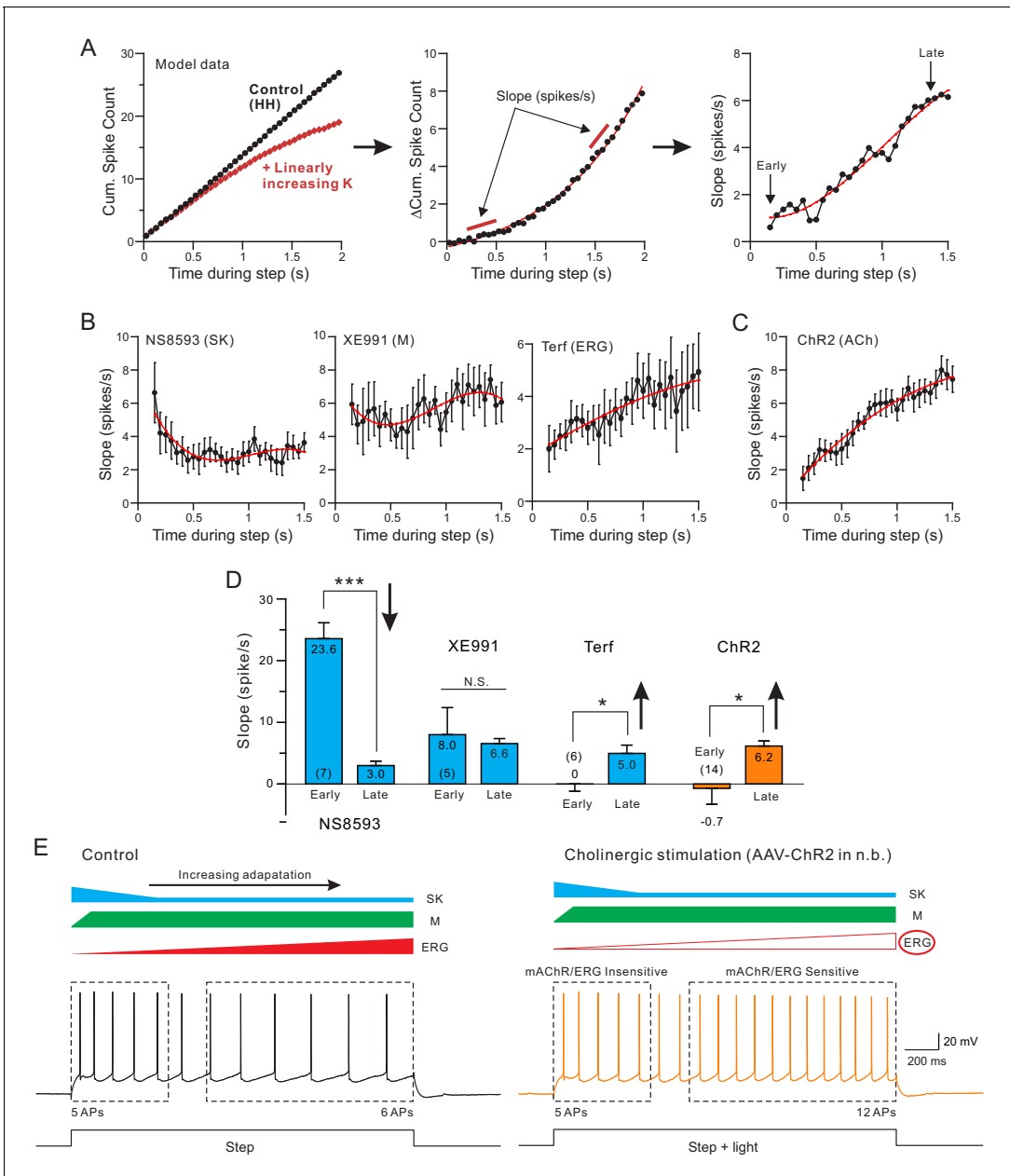

**Figure 6.** Both terfenadine and endogenous ACh reduce a current responsible for late-phase spike frequency adaptation. (**A**) demonstration of method used to define kinetics of currents responsible for SFA. From a standard pair of cumulative spike count plots (left panel, responses from a computer simulation with and without a linearly-increasing K⁺ current), a difference cumulative spike count function is computed (middle panel). Finally, a series of slope estimates, representing the instantaneous rate of SFA, is generated at different times during the step response. Slopes at the 'Early' and the 'Late' phases of the response are quantified in D. (**B**) Estimates of the kinetics of SFA modulated by the same three K⁺ channel blockers assayed in *Figure 5*, revealing the declining functional effect of SK current (left panel) and the increasing functional effect of Terf-sensitive ERG current during the same type of 2 s duration depolarizing step responses. (**C**) Rate of SFA increases during step responses when endogenous ACh is released. (**D**) Summary of the initial and final rates of SFA. Blockade of SK channels with NS8593 decreased the rate of SFA: ***p=0.000249, T = 7.714, while both Terf and ChR2-mediated ACh release increase the rate of SFA: * Terf: p=0.0432, T = 2.692; * ChR2 p=0.0471, T = 2.192; XE991 n.s., p=0.923, T = 0.102; all paired t-tests. Experiments using Terf and XE991 were conducted using an internal solution with strong Ca²⁺ buffering (10 mM BAPTA) to attenuate SK-mediated effect. (**E**) Schematic diagram of the kinetics of the main K⁺ currents hypothesized to underlie SFA under control conditions (left: SK, blue; M, green; ERG, red) and following mAChR activation (right) with attenuated ERG and reduced late-phase SFA. The spiking response during the early phase of the step (left dashed box) was insensitive to mAChR modulation and was likely mediated by M or SK current. In contrast, neuronal excitability was selectively increased in the late phase (right dashed box), likely by mAChR-mediated attenuation of ERG. Intracellular responses duplicated from *Figure 1B*.

*Figure 6 continued on next page*

*Figure 6 continued*

DOI: https://doi.org/10.7554/eLife.44954.015

The following figure supplement is available for figure 6:

**Figure supplement 1.** Rate of spike frequency adaptation was increased following ERG blockade with Terf in experiments with weak Ca$^{2+}$ buffering.

DOI: https://doi.org/10.7554/eLife.44954.016

themselves (*Wolfart and Roeper, 2002*) or by changes in the signaling pathways governing the SK current. The effectiveness of adaptation mediated by M current, by contrast, was relatively constant throughout the step (middle panel in *Figure 6B*). As with the simulation example plot with linearly-increasing K$^+$ current, the rate of SFA increased during the step both when ERG currents were attenuated by Terf with high BAPTA internal (right panel in *Figure 6B*) and when ChR2 stimulation released ACh (*Figure 6C*). The increasing rate of SFA in these experiments suggests that the functional impact of ERG K$^+$ currents increases during the step response, providing a final barrier to maintaining the initial high firing rate throughout the step response. In *Figure 6D* we compare the SFA rates at the beginning and end of the step response and find that the adaptation mediated by SK NS8593-sensitive SK channels diminished during the step response while, conversely, the adaptation mediated by ERG channels increased. Similar to Terf, ChR2-stimulated ACh release modulated the SFA rate preferentially during the late phase of the step response. The similarity between the SFA dynamics associated with release of endogenous ACh (orange bars in *Figure 6D*) and Terf (right-most blue bars in *Figure 6D*) provides additional evidence that under our experimental conditions, ACh enhanced neuronal excitability primarily by attenuating ERG current. We found a similar increase in SFA rate (slope) when ERG was blocked with Terf in a low EGTA internal solution (*Figure 6—figure supplement 1*).

Together, these results suggest that the specificity of endogenous ACh for attenuating the slowly developing ERG K$^+$ current likely explains pattern of hyperexcitability we observed in our experiments. As illustrated in the diagrams in *Figure 6E*, left, normal SFA appears to involve at least three K$^+$ subtypes that turn on in a stereotyped temporal sequence. The initial phase of SFA is controlled primarily by SK channels (blue) and the M current (green). Among these two K$^+$ subtypes initially recruited, our results suggest that SK begins to slow firing before the M current because only SK modulators could influence the first spike time. Later in the response the influence of SK diminishes while SFA mediated by M current is maintained. Finally, a third K$^+$ subtype slowly develops (ERG, red) that further slows firing. In response to physiological (5–15 Hz) firing rates, ERG-mediated SFA begins after approximately 500 ms. While all three K$^+$ currents that contribute to SFA can be modulated by artificial mAChR agonists, our results indicate that endogenous ACh selectively attenuates the ERG mediated component. Because of this specificity, cholinergic stimulation is able to selectively modulate long-lasting responses while leaving responses to brief stimuli unaltered (e.g., the same number of spikes in the first dashed box in *Figure 6E*).

## Endogenous ACh enables persistent firing

Our results thus far demonstrated that ChR2-stimulated ACh release both enhances responses to depolarizing stimuli and promotes afterdepolarization responses. Both effects parallel previous results found using bath application of muscarinic receptor agonists such as CCh (e.g., *Knauer et al., 2013* in hippocampus and *Cui and Strowbridge, 2018*, *Rahman and Berger, 2011*, in neocortex). Another key result from these previous bath application studies was the demonstration of cell-autonomous stimulus-triggered persistent firing, often observed when neurons were depolarized slightly from their resting membrane potential.

We found similar persistent firing modes following the release of endogenous ACh. As shown in *Figure 7A*, slightly depolarizing L5 pyramidal cells revealed persistent firing in response to coincident step + light stimuli. Repeating the same step + light stimulation at −70 mV generated only an ADP response following the step (*Figure 7A*, left). No persistent firing occurred when the step depolarization was presented alone (black trace in *Figure 7A*, right) or steps were paired with preceding or subsequent ChR2 stimulation (i.e., not coincident, c.f., *Figure 1H*; *Figure 7—figure supplement 1A-B*). Persistent firing triggered by coincident ChR2 light/step stimulation required mAChR activation since it was abolished by atropine (*Figure 7B*). In addition to testing the broad-

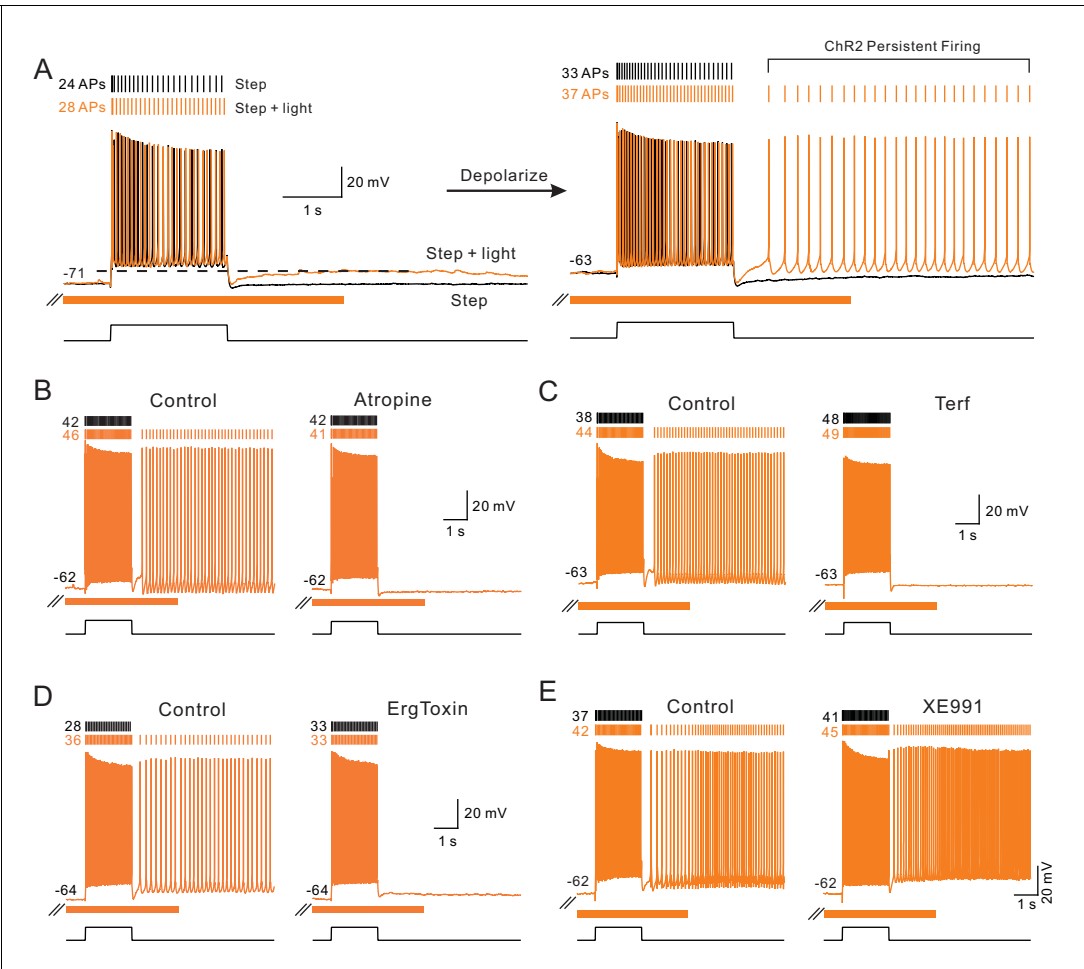

**Figure 7.** Endogenous ACh enables intrinsic persistent firing by decreasing ERG current. (**A**) ChR2-evoked ACh release enables a long-duration afterdepolarization (ADP) following depolarizing steps that remained subthreshold when tested at ~−70 mV (orange trace in left panel). No ADP is evident in interleaved step-only trials (black trace). Depolarizing the same pyramidal cell by 8 mV allowed the post-step ADP to trigger persistent firing when ACh was released via ChR2 light stimulation (orange trace in right panel). (**B**) The muscarinic receptor antagonist atropine (10 $\mu$M) abolishes ChR2-mediated persistent firing and the increase in the number of APs evoked during the step stimulus. (**C**) The ERG antagonist terfenadine (Terf, 10 $\mu$M) abolishes ChR2-mediated persistent firing. (**D**) Persistent firing mediated by ChR2 stimulation also was abolished by ErgToxin1 (50 nM). (**E**) XE991 (10 $\mu$M), an M current antagonist, does not block ChR2-mediated persistent firing.

DOI: https://doi.org/10.7554/eLife.44954.017

The following figure supplements are available for figure 7:

**Figure supplement 1.** Additional pharmacological properties of ChR2-mediated persistent activity.

DOI: https://doi.org/10.7554/eLife.44954.019

**Figure supplement 2.** Persistent activity requires coincident step depolarization and ChR2 light stimulation.

DOI: https://doi.org/10.7554/eLife.44954.018

spectrum muscarinic receptor antagonist atropine, we also tested two subtype specific blockers. In 3/3 experiments, the m1 class muscarinic receptor antagonist pirenzepine (10 $\mu$M; *Freedman et al., 1988*) abolished persistent firing mediated by ChR2 stimulation (example recordings in *Figure 7— figure supplement 1A*). By contrast, AF-DX 116 (10 $\mu$M), an m2 class receptor antagonist, failed to abolish ChR2 mediated persistent firing in 3/3 experiments (*Figure 7—figure supplement 1B*), providing evidence that endogenous ACh modulates excitability and gates persistent firing in these experiments via actions on m1 class muscarinic receptors.

Since we demonstrated above that ERG blockers abolished the increase in excitability to depolarizing stimuli, we next asked if the same ERG antagonists affected ChR2-gated persistent firing. As shown in *Figure 7*, persistent firing was completely blocked by both Terf (10 $\mu$M, *Figure 7C*) and

ErgToxin1 (50 nM, *Figure 7D*). We also found that third ERG blocker employed in the experiments described above (E-4031, 10 $\mu$M) similarly blocked persistent firing (*Figure 7—figure supplement 1C*). By contrast, persistent firing was not blocked by antagonists of the M current (XE991, 10 $\mu$M, *Figure 7E*) and SK channels (NS8593, 10 $\mu$M). Results from these pharmacological studies are summarized below and compared with parallel studies of persistent firing following bath CCh application.

The persistent firing behavior we find following coincident release of endogenous ACh and depolarization was similar to that described in our previous study of L5 neocortical pyramidal cells exposed to bath CCh (*Cui and Strowbridge, 2018*). As shown in *Figure 8A*, muscarinic receptors activation with either endogenous ACh (orange) or CCh (green) triggered persistent firing at similar frequencies (~5 Hz; *Figure 8B*). Among our population of L5 pyramidal cells in which ChR2 stimulation increased the number of spikes evoked during step stimuli, a majority (73%) had persistent firing that lasted at least 2 s following the depolarizing stimuli. In the same cell type, bath CCh also enabled persistent firing in large majority (87%) of cells tested (*Figure 8C*).

A recent study from our group assayed dynamic changes in input resistance associated with persistent activity and found a large increase in input resistance underlying persistent firing in CCh (*Cui and Strowbridge, 2018*). Using the same method based on trains of brief hyperpolarizing steps (*Figure 8D*, top) corrected by the effect of depolarization (using Rin vs membrane potential calibration data, *Figure 8—figure supplement 1*), we find a similar increase in input resistance during persistent firing which was maximal 2–3 s following step offset (*Figure 8D*, middle). The post-step increase in input resistance decayed with similar kinetics in both ChR2-ACh and bath CCh conditions (*Figure 8D*, bottom).

A reduction in the leak $K^+$ current mediated by ERG appears to facilitate persistent firing in CCh (*Cui and Strowbridge, 2018*). Because of its slow activation and recovery kinetics (*Cui and Strowbridge, 2018*, *Niculescu et al., 2013*), pyramidal cells appear to experience a post-stimulus increase excitability until the steady-state leak $K^+$ conductance mediated by ERG is restored. When we compared the ability of different $K^+$ channel antagonists to block persistent firing gated by either endogenous ACh (via ChR2 stimulation) of bath CCh, we found similar results. All three structurally-diverse ERG antagonists tested (Terf, E-4031 and ErgToxin) abolished persistent firing in ChR2 (*Figure 8E*) and in our previously published (*Cui and Strowbridge, 2018*) data set using bath CCh (*Figure 8F*). In each ChR2 experiment, we verified that light stimulation reliably triggered persistent firing over at least three different trials in each cell. The absence of persistent firing upon treatment with ERG blockers was unlikely to reflect an attenuated depolarizing stimulus as the number of APs evoked by the step depolarization increased with ERG blockade (number of step-evoked APs indicated next to current clamp voltage responses in *Figure 7C–D*). The blockade of persistent firing by ERG antagonists also was unlikely to reflect rundown in the ChR2 experiments because we could reliably trigger persistent firing for over one hour (>10 trials; N = 3 mock drug application experiments). Persistent firing following both ChR2 stimulation and bath CCh application was resistant to block by antagonists of the M current (XE991) and SK channels (NS8593). Together, these results indicate that light-stimulated release of endogenous ACh can selectively reduce $K^+$ currents mediated by ERG channels, leading to both enhanced neuronal excitability and, at more depolarized membrane potentials, intrinsic persistent firing.

## Network consequences of selective modulation of ERG by ACh

Both attention and muscarinic activation can enhance cellular and network excitability in ways that increase signal-to-noise ratio (SNR; *Cohen and Maunsell, 2009*, *Sato et al., 1987*, *Ballinger et al., 2016*, *McAdams and Maunsell, 1999*, *Zinke et al., 2006*, *Picciotto et al., 2012*). In the final part of this study, we asked if network SNR and related changes in network correlation could have, at least in part, a biophysical basis–can previously observed changes in the statistical properties of ensemble firing patterns during periods of heightened attention relate to ACh-triggered reductions in SFA? In principle, increasing excitability does not necessarily modulate SNR (e.g., if both signal and noise change in tandem; *Figure 9A*) while in vivo, ACh often increases both excitability and SNR (*Kang et al., 2014*, *Hasselmo and Sarter, 2011*, *Picciotto et al., 2012*, *Benarroch, 2010*). The potential contribution of intrinsic properties to these network changes have not been established experimentally.

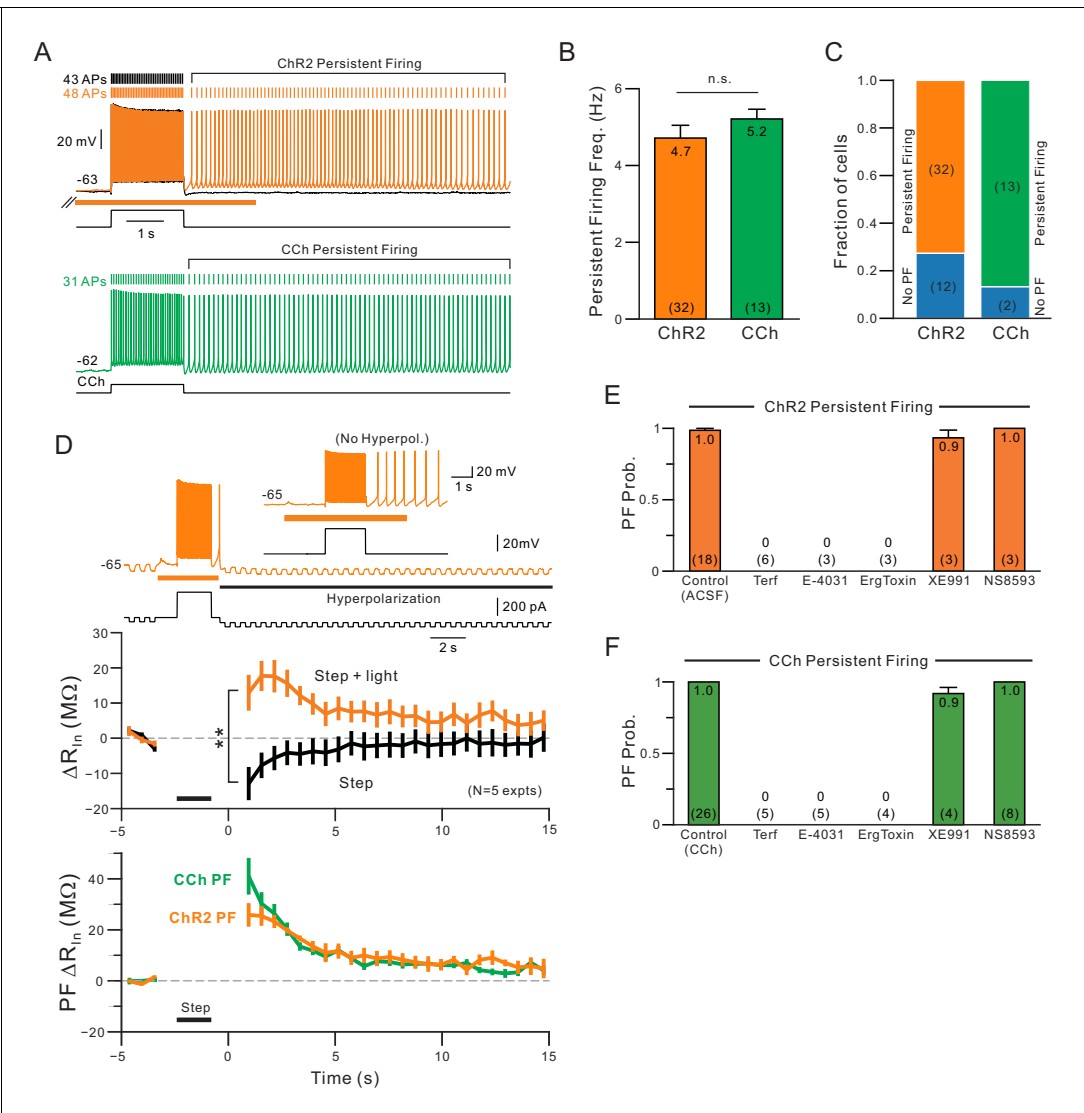

**Figure 8.** Comparison of persistent firing properties and pharmacology with endogenous ACh release and bath application of carbachol. (**A**) Comparison of persistent firing evoked by depolarizing steps combined with ChR2-mediated ACh release (orange trace, top) and with bath application of carbachol (CCh, 2 $\mu$M, green, bottom trace). Responses from different cells but at same vertical and time scales. (**B**) Summary of average persistent firing frequency computed over 9 s with ChR2-mediated ACh release (orange) and bath CCh (green). (**C**) Summary of proportion of pyramidal cells tested with ChR2 stimulation that had persistent firing at ~−63 mV (left) and with bath CCh at the same membrane potential (right). (**D**) Endogenous ACh-mediated persistent firing is associated with an increase in input resistance. Top traces illustrate protocol used to assay input resistance, adopted from a related study (**Cui and Strowbridge, 2018**), where persistent firing that would normally occur is suppressed by hyperpolarizing the membrane potential; example response from the same neuron without added post-stimulus hyperpolarization shown in inset. Middle plots show summary of input resistance change from pre-stimulus levels at each time step in control conditions (step-only, black plot) and following ACh release (step + light, orange plot). Immediately following the depolarizing stimulus, input resistance decreased relative to baseline in control while it increased when ACh was present. **p=0.00752, T = 4.994, paired t-test. Calibration routine used to establish baseline input resistance at different membrane potentials was shown in the supplement for this figure. Bottom, plot of persistent firing associated increase in input resistance in ChR2 stimulation experiments (orange plot, difference between orange and black plots in middle panel) and bath CCh experiments (green plot, difference in input resistance plots before and after CCh application). Timing of 2 s depolarizing step indicated by black horizontal line in middle and bottom panels. (**E**) Summary of probability of triggering persistent firing in ChR2 experiments following treatment with ERG blockers (terfenadine, E-4031 and ErgToxin1) and antagonists for other K+ channels (XE991 for M current and NS8593 for SK channels). (**F**) Parallel summary plot for experiments conducted with bath CCh (and without ChR2 stimulation). Experiments summarized in (**E-F**) were conducted on a subset of experiments where depolarizing step stimuli triggered persistent firing under baseline conditions (either with ChR2 light stimulation or bath CCh).

DOI: https://doi.org/10.7554/eLife.44954.020

The following figure supplement is available for figure 8:

*Figure 8 continued on next page*

*Figure 8 continued*

**Figure supplement 1.** Calibration for dynamic Rin measurements.
DOI: https://doi.org/10.7554/eLife.44954.021

Using similar methods to estimate SNR as employed in previous extracellular unit recordings in vivo (*Cohen and Maunsell, 2009*, *McAdams and Maunsell, 1999*), we find that ChR2 stimulation increased SNR across virtual ensembles of L5 pyramidal cells (*Figure 9B–C*). We also found that reducing ERG-like linearly increasing K$^+$ currents enhanced network SNR in our computational models (*Figure 9—figure supplement 1*). The increase in SNR in our experiments was due to enhancements of signal while the noise component was not changed (*Figure 9D–E*). These results demonstrate that cholinergic modulation of biophysical properties of neocortical pyramidal cells can explain the selective enhancement of signal over noise components commonly observed following cholinergic stimulation in vivo.

Regulation of temporal patterning by late-developing K$^+$ currents such as ERG also provides potential explanations for another common finding in in vivo studies of attention and ACh

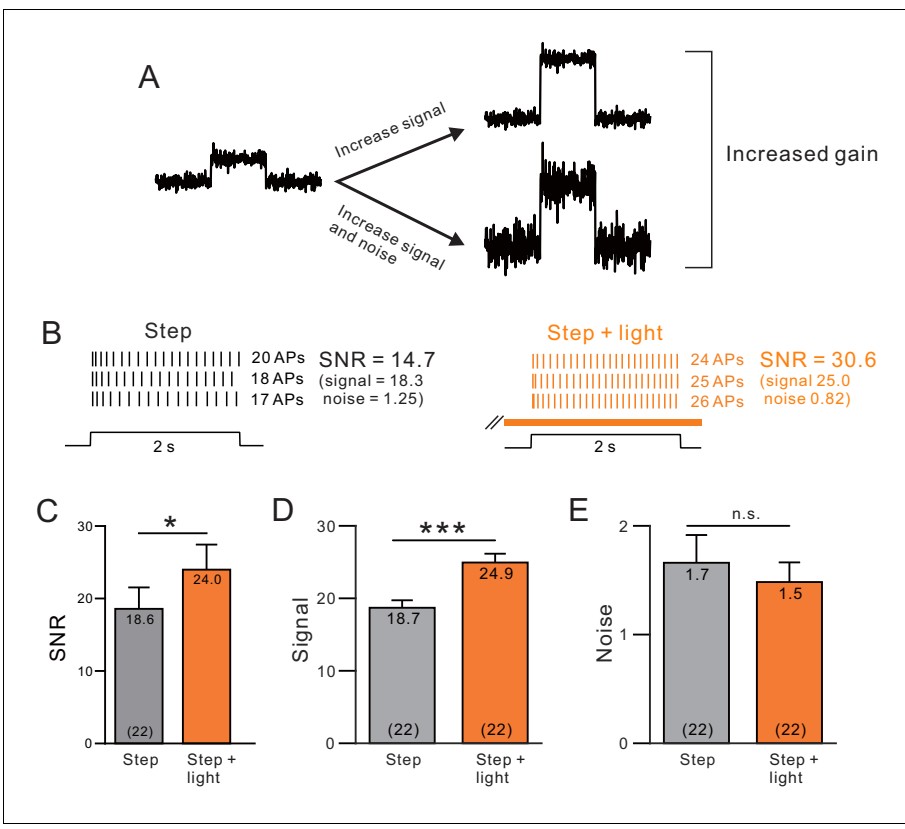

**Figure 9.** A, Endogenous ACh enhances signal-to-noise ratios by selectively increasing the signal component. (**A**) Diagram of gain enhancement with (top) and without (bottom) a corresponding increase in SNR. (**B**) An example cell showing average spiking responses and signal-to-noise ratio (SNR) were increased by light stimulated ACh release. (**C**) Summary of SNR estimates from 22 ChR2-stimulated cells. ChR2 mediated ACh release significantly increased SNR (* ChR2 step vs. step + light: p=0.0343, T = 2.263). Isolating the signal (**D**) and noise (**E**, SD of spike count) components revealed a selective enhancement of the signal (mean spike count). *** P (signal) = 2.25E-9; T = 9.914; P (noise) = 0.360; T = 0.935; both paired t-tests.
DOI: https://doi.org/10.7554/eLife.44954.022

The following figure supplement is available for figure 9:

**Figure supplement 1.** Reduction in signal-to-noise ratios with ERG-like K$^+$ currents in a computational network model.
DOI: https://doi.org/10.7554/eLife.44954.023

stimulation: a reduction in temporal signal correlation (*Steinmetz et al., 2000*, *Goard and Dan, 2009*). While the underlying mechanism responsible for reduced correlation among neural ensembles during behavioral tasks is not known, one potential explanation may be an attenuation of the correlation normally imparted on discharges by intrinsic physiological currents such as those mediating SFA. As illustrated in the diagram in *Figure 10A*, uncorrelated inputs to two neurons can generate correlated spike discharges if both neurons have SFA intrinsic properties that dampen the late phases of the responses. As demonstrated in many previous studies and the present report, the intrinsic properties in most neocortical pyramidal cells normally function to adapt firing rates and therefore could create correlated firing rates within ensembles. Since we find endogenous ACh functions to attenuate this SFA, one potential consequence of cholinergic stimulation could be to reduce network signal correlations. To our knowledge, this hypothesis has not been tested experimentally, though multiple computation studies (*Crook et al., 1998*, *Ermentrout et al., 2001*, *Wang et al., 2014a*) have examined whether adaptation-based models can explain changes in signal correlation. It is not known, therefore, whether the similarity in adaptation rates between cells would overcome the normal variability in intrinsic properties between individual pyramidal cells to generate reliable signal correlations.

We assayed correlation in spike discharges by computing a single correlation coefficient from each pair of neurons tested under the same condition (eg., step only or step + light) and then repeated that process within virtual ensembles composed of the entire ChR2 data set. In each pairwise test, we computed the correlation (Pearson's R) between the spike counts in corresponding time bins within the step response in both cells (*Figure 10B–C*). While information about the time sequence of each bin was discarded during this analysis, the resulting correlation coefficient still reflected similarity in temporal patterning as long as corresponding time bins were compared in each cell. Randomizing the order of time bins abolished correlations normally present when two adapting discharges are compared (see right example pairwise comparison in *Figure 10C*). This method represents a robust approach for assessing whether two cells share a similar temporal pattern of firing rate modulation without necessitating more complex (and higher dimensional) methods that explicitly include time information. Our approach has been used in previous in vivo studies to assay modulation of signal correlation within virtual neural networks by ACh (*Goard and Dan, 2009*, *Minces et al., 2017*).

Across 24 light-stimulated ACh release experiments, we found significantly less temporal signal correlation in responses acquired during light stimulation than in control conditions. This reduction in temporal signal correlations is reflected by the majority of experiments showing less correlation when ACh is released than in control (step-only) conditions in the plot shown in *Figure 10D* which replicated a previously published in vivo analysis (*Goard and Dan, 2009*). In this analysis, we computed the average signal correlation from one cell compared with the 23 other cells in the data set under both step only and step + light conditions and then repeated the process for each cell. If ACh had no effect on signal correlation, most points would lie near the equal correlation red diagonal line instead of below the diagonal. Considering all 276 pairwise comparisons within our population of 24 experiments, ChR2 stimulation reduced the average signal correlation by 22% (from 0.27 to 0.21; *Figure 10E*). In vivo, many intrinsic and synaptic factors can affect signal correlation. In our brain slice experiments, by contrast, the attenuated signal correlation likely reflects the reduced SFA described in single cell analyses above. As SFA is reduced by ACh, firing becomes more tonic and correlations between discharges in pairs of cells diminishes. Randomizing the order time bins compared in each cell abolished the signal correlation in both control and ChR2 stimulation conditions (right bars in *Figure 10E*).

The reduction in signal correlation with ACh release developed over the neuronal response, as expected for the selective attenuation of a late component of SFA by ACh. Signal correlations calculated from the initial few time bins, representing only the early phase of the response, were not affected by ChR2 stimulation while network correlations were reduced when the entire response was analyzed in the same set of experiments. As shown in *Figure 10F*, ChR2 stimulation significantly reduced signal correlation (to $p<0.01$) only when responses longer than 625 ms were analyzed. The similarity in response duration required for ACh-mediated signal correlation attenuation (~625 ms) and the onset of ERG-mediated SFA (~575 ms from experiment shown in *Figure 5C*) suggests that the attenuated signal correlation could reflect the reduction in the late phase of SFA across the network. The same time-dependent pattern in signal correlation change was recapitulated in

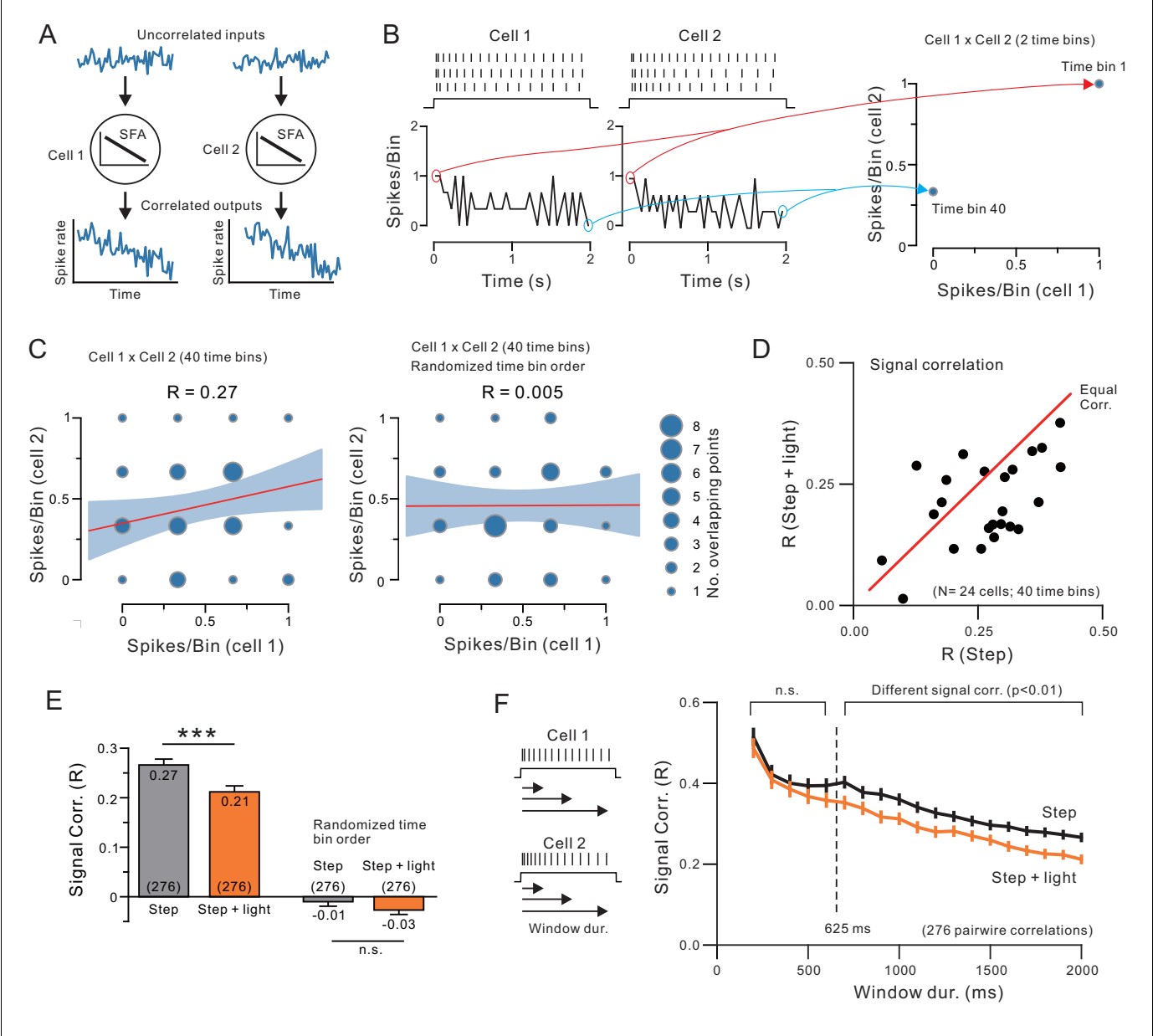

**Figure 10.** Endogenous ACh reduces signal correlation by modulating intrinsic conductances. (**A**) Diagram illustrating how similar SFA can force two neurons to have correlated output discharges. (**B**) Example spiking responses to depolarizing steps recorded in two neocortical pyramidal cells and plots of spike count per 50 ms time bin (left). Plot on right illustrates partial correlation matrix computed from only two time bins (the first and last) from the example discharges. (**C**) Signal correlation was calculated by computing the Pearson's correlation (R) between spike counts in all 40 time bins in each pair of neurons (left; linear regression and 95%ile confidence interval superimposed on correlation matrix). No signal correlation is generated in the same two-cell example comparison when time bin order is randomized (right). (**D**) Plot of signal correlation under control and ACh-stimulation in 24 experiments. Equal correlation levels indicated by red diagonal line. (**E**) Summary plot of signal correlation from all pairwise comparisons possible within virtual network of 24 neurons. ChR2 light stimulation significantly reduced signal correlation; ***p<0.0002, T = 3.932, paired t-test. Randomizing the time bin order abolished the signal correlation present in both control and ChR2 light stimulation conditions. (**F**) Plot of signal correlation vs. response duration in control (grey) and ACh-stimulated conditions (orange). Same dataset as E. Corresponding pairs of control and ACh-stimulated points differ significantly at response durations > 625 ms; at 650 ms: p=0.00224, T = 3.085, paired t-test. Diagram illustrating analysis protocol shown on left.

DOI: https://doi.org/10.7554/eLife.44954.024

The following figure supplement is available for figure 10:

**Figure supplement 1.** Computation models incorporating an ERG-like K⁺ conductance that increases during depolarizing step responses reproduces changes in signal correlation.

*Figure 10 continued on next page*

*Figure 10 continued*

DOI: https://doi.org/10.7554/eLife.44954.025

computational simulations when the ERG-like linearly-increasing K⁺ current was reduced (***Figure 10—figure supplement 1A,c.f.*** experimental ***Figure 10F***). The decrease in signal correlations in computer simulations where the ERG-like K⁺ current was attenuated was apparent with low to moderate discharge frequencies, including the standardized ~ 10 Hz discharges examined in this study. Signal correlations between neurons with higher frequency discharges were modulated less by the linearly increasing K⁺ current, though they remained statistically different throughout the range of stimulus amplitudes tested in our model (***Figure 10—figure supplement 1B***). These computational simulation results suggest that the coordinated reductions in ERG-like conductances across neurons within a cortical network are sufficient to reduce ensemble signal correlation and this effect was robust over a wide range of discharge frequencies.

The reduction in signal correlation also may underlie network desynchronization, since in our experiments, ACh appears to attenuate the slow temporal pattern imparted by each neuron's intrinsic physiology. These results suggest that a component of network desynchronization may reflect attenuation of the 'baseline' temporal patterning imparted to most neuronal discharges by SFA. Previous work has demonstrated the association between signal correlation and desynchronization in local field potential commonly observed in vivo upon NB stimulation (***Metherate and Ashe, 1993***, ***Kalmbach et al., 2012***, ***Pinto et al., 2013***). Dan and colleagues (***Goard and Dan, 2009***) assayed with both local field potential and unit recordings in rat visual cortex while presenting natural stimuli. Local field potential was desynchronized and signal correlation was also reduced upon NB electrical stimulation. These results suggest that similar biophysical modulation by ACh may be responsible for both LFP desynchronization and reduction in signal correlation. While many diverse mechanisms probably contribute to network desynchronization in vivo, our results highlight the potentially important role of changes in intrinsic physiology for effecting the changes in network statistical structure commonly observed during periods of heightened attention and NB stimulation, situations in which neocortical ACh release is enhanced (***Richardson and DeLong, 1986***, ***Himmelheber et al., 2000***, ***Bastiaansen et al., 2001***, ***Arnold et al., 2002***).

## Discussion

We make three primary conclusions in this study. First, that the normal firing rate adaptation commonly observed in L5 pyramidal cells likely reflects the effects of at least three different K⁺ currents that are recruited in a stereotyped order (first SK, then M, and then finally ERG). While previous reports have studied the isolated contributions from these currents, especially SK (***Bond et al., 2004***, ***Pedarzani et al., 2005***, ***Ji et al., 2009***, ***Yen et al., 1999***, ***Lin et al., 2010***), we believe this study is the first to demonstrate the temporal sequence of the three primary currents involved in SFA as well as the delayed contribution of ERG to adapting pyramidal cell firing rates.

Second, we find that coincident release of endogenous ACh during depolarizing stimuli selectively decreases the late phase of SFA by reducing ERG by activating muscarinic receptors. The preferential modulation of ERG by endogenous ACh enables cholinergic stimulation to selectively enhance responses to long-lasting stimuli while leaving responses to brief stimuli unaffected. The preferential modulation of a leak K⁺ current with slow kinetics also explains the ability of cholinergic stimulation to enable persistent firing following depolarizing steps in quiescent neurons, a phenomenon widely reported following bath application of muscarinic receptor agonists (***Egorov et al., 2002***, ***Rahman and Berger, 2011***, ***Knauer et al., 2013***, ***Jochems and Yoshida, 2013***, ***Cui and Strowbridge, 2018***) but not reported previously following ChR2-stimulated release of endogenous ACh. Two phenomena associated with muscarinic cholinergic modulation of neuronal excitability (enhanced neuronal excitability and enabling persistent firing that outlasts the triggering depolarizing stimuli) are both mediated by attenuation of ERG currents.

And finally, we determined that the same ERG modulation mechanism that mediates the late phase of SFA can explain previously reported changes in the signal-to-noise ratio and correlation properties assayed in network recordings in vivo. Together, these findings reveal that cholinergic inputs can modulate a specific subclass of K⁺ currents out of a larger constellation of potential

candidate currents, leading to distinctive changes in intrinsic physiology that can potentially explain many features of attention-related working memory tasks.

## Modulation of spike frequency adaptation by ACh

The intrinsic currents underlying SFA have been studied extensively in many cell types with most reports focusing on SK-mediated AHP and the M current. The $Ca^{2+}$-dependent AHP response is assayed soon after a step-evoked burst of APs (*Peron and Gabbiani, 2009*, *Ha and Cheong, 2017*) but is presumably active throughout spiking responses, functioning to slow firing rates (*Storm, 1989*). Multiple studies implicated calcium-activated small-conductance $K^+$ (SK) channels in mediating at least partially AHP and SFA (*Bond et al., 2004*, *Pedarzani et al., 2005*, *Ji et al., 2009*, *Yen et al., 1999*, *Lin et al., 2010*). While SK-mediated adaptation explains the common finding that increasing intracellular $Ca^{2+}$ buffering reduces SFA, it cannot be the sole mechanism as some adaptation remains following even strong $Ca^{2+}$ chelation (*Madison and Nicoll, 1984*, *Storm, 1989*). Previous studies also have implicated $K^+$ currents active near rest in contributing to SFA, including the M current (*Aiken et al., 1995*, *Gu et al., 2005*) and ERG (*Chiesa et al., 1997*, *Sacco et al., 2003*, *Pessia et al., 2008*). While there are abundant data available about both SK and leak $K^+$ currents in isolation (especially within expression systems), little is known about how these different currents function together to effect adaptation in response to physiological discharges.

Many classic studies approached the question of which $K^+$ currents contribute to SFA by assaying afterhyperpolarizations following different duration depolarizing steps. Crill and colleagues (*Schwindt et al., 1988a*, *Spain et al., 1987*, *Schwindt et al., 1988*, *Schwindt et al., 1988b*) identified three primary classes of AHPs based on kinetic differences, termed fast, medium and slow. Based on more recent studies, big-conductance $Ca^{2+}$-activated $K^+$ channels (BK) channels like contribute to the fastest AHP (*Gu et al., 2005*) while multiple currents appear contribute to the medium AHP, including SK channels (*Schwindt et al., 1988a*), I-H (*Spain et al., 1987*, *Schwindt et al., 1988*, *Gu et al., 2005*) and the M current (*Storm, 1989*, *Gu et al., 2005*). The identity of the $K^+$ current mediating the slow AHP has remained elusive. Based on results in the present study, ERG likely contributes to the slow AHP described in these prior studies. Previous work (*Schwindt et al., 1988b*) demonstrated the sensitivity of slow AHP to muscarinic receptor activation which evoked a coordinated reduction in slow AHP amplitude and SFA, paralleling results we obtained following ERG attenuation in the present study.

Our results implicated SK and two leak $K^+$ currents in mediating SFA and, for the first time, defined their temporal activation sequence in response to standardized neocortical discharges (~10 Hz firing; *Cowan and Wilson, 1994*, *Sanchez-Vives and McCormick, 2000*): SK, then M, and finally ERG. Three lines of evidence support the hypothesis that the initial phase of SFA is mediated by SK current. First, only SK modulators affected first spike timing. Blockers of M and ERG $K^+$ currents increased excitability but did not affect the timing of the initial spike. Second, the ability of SK modulators to bidirectionally modulate first spike timing suggested that this $K^+$ current could be triggered by VGCC-mediated depolarizations (and thus $Ca^{2+}$ influxes) that were subthreshold for AP generation. We confirmed that prediction by demonstrating that SK inhibitors reduced the AHP following brief subthreshold depolarizations. Finally, bifurcations in cumulative SFA plots revealed with SK inhibitors occurred early in the step response and before the bifurcation time generated by M and ERG blockers. We deduced the order that leak $K^+$ currents contribute to SFA by comparing bifurcation time following pharmacological blockade of either the M current or ERG (the latter tested under two different $Ca^{2+}$ buffering conditions). These results suggest that following the initial phase of SK-mediated adaptation, the M current and then ERG further slow firing. In addition to different onset latencies, the $K^+$ currents that mediate SFA appear to have different kinetics with SK currents reducing soon after their maximal functional effect while M currents remained constant and the adaptation mediated by ERG currents increased with time.

While we are not aware of previous studies that attempted to determine the temporal sequence of currents mediating SFA in L5 neocortical pyramidal cells, the activation order we find is consistent with the established biophysical properties of these channels. Previous work demonstrated that SK current can be rapidly activated by subthreshold T-type $Ca^{2+}$ current (*Wolfart and Roeper, 2002*, *Matschke et al., 2018*). This and other prior work (*Gu et al., 2005*, *Vandael et al., 2012*) is consistent with our hypothesis that SK could be active even before the first spike within the firing responses we tested (at ~ 10 Hz) and, therefore, could delay the onset of the first spike. M currents

are typically regarded as slow (although not as slow as ERG) K$^+$ current with activation time constant of ~ 100 ms (*Prole et al., 2003*)–explaining the lack of first spike time modulation with XE991–while ERG is even slower with activation time constant of 1 ~ 2 s (*Wang et al., 1997a*). Consistent with our results, several previous studies also failed to find the initial first spike latency affected following M current blockade with either linopirdine (*Aiken et al., 1995*) or XE991 (*Gu et al., 2005*).

## Selective modulation of ERG by endogenous ACh

One functional consequence of the kinetically diverse group of K$^+$ currents that mediate to SFA is that neuromodulators can shape neuronal (and likely ensemble) responses by selectively regulating one constituent current. All three K$^+$ currents we find contribute to SFA in L5 neocortical pyramidal cells (SK, M and ERG) have been reported to be attenuated following mAChR activation (*Buchanan et al., 2010*, *Giessel and Sabatini, 2010*, *Suh and Hille, 2007*, *Hirdes et al., 2004*, *Shapiro et al., 2000*). However, ERG appears to be the primary target of endogenous ACh released via ChR2 light stimulation of NB axons. The selectivity for ERG is suggested by four findings. First, the bifurcation time assayed in response to the standardized discharge stimulus was closest to the bifurcation time following pharmacological ERG blockade and much more delayed than following blockade of either SK or M currents. Second, the insensitivity of bifurcation time following ACh release to variation in the onset time of ChR2 stimulation likely means that the target of modulation has slow kinetics (rather than slow intracellular signaling cascades triggered by ACh). The established slow kinetics of ERG currents in a variety of neurons (*Wang et al., 1997b*, *Cui and Strowbridge, 2018*) closely matches the bifurcation time we find following ACh release. Third, endogenous ACh did not affect the first spike time, a functional signature of manipulations that affect SK but not M or ERG currents. And finally, instantaneous SFA kinetics (adaptation slope) following endogenous ACh release matched the pattern we find following ERG blockade (apparent current increase during discharge) and differed from the patterns we defined following blockade of either SK or M current.

While several previous studies described increased neuronal excitability following ChR2 stimulated ACh release (*Joshi et al., 2016*, *Baker et al., 2018*, *Hedrick and Waters, 2015*), we believe this is the first report that focuses on the continuous release of endogenous ACh throughout neocortical neuron discharges. Most of these previous studies found that optogenetic stimulation alone increased spontaneous firing (*Baker et al., 2018* and NB electrical stimulation in *Metherate et al., 1992*) or promoting spontaneous firing in near-threshold neurons (*Joshi et al., 2016*, *Hedrick and Waters, 2015*) through activation of mAChRs. In our experiments, releasing ACh alone via ChR2 stimulation failed to modulate excitability in quiescent pyramidal cells while increases in excitability required coincident ACh release with depolarizing stimuli. Baker et al. (*Baker et al., 2018*) addressed coincident ACh and depolarizations by optogenetically stimulated cortical cholinergic fibers with a brief 5 ms pulse of light immediately preceding a 1500 ms depolarizing step and demonstrated increased excitability which was partially reduced by M current blockers. (This study did not test ERG blockers.) However, none of the studies mentioned above activated cholinergic fibers throughout the discharge, recapitulating the prolonged firing patterns NB neurons generate during attentive behaviors (*Sarter et al., 2009*) and which may be necessary to reveal modulation of ERG-mediated SFA.

Additional studies will be required to determine whether ERG selectively regulates the late phase of SFA and persistent firing in other neocortical cell types, such as intrinsic bursting and superficial pyramidal cells. In prefrontal cortex, L5 pyramidal cells can be classified into PT or IT subtypes based on their projection targets (*Dembrow et al., 2010*, *Dembrow and Johnston, 2014*). Based on previously reported intrinsic physiological properties of prefrontal cortical neurons, the L5 pyramidal cells we recorded from more closely resemble PT cells, reflecting their sensitivity to mAChR activation, high sag ratio and similar input resistance as well as similar AP threshold and width properties. However, prefrontal PT do not show the marked SFA we find in TeA L5 regular spiking neurons. Additional studies will be required to determine if this difference reflects the different internal solutions used in these studies or whether TeA neocortex does not contain analogous cell types to the PT/IT subclasses defined in the prefrontal cortex.

The central role we find for ERG modulation by endogenous ACh also predicts that ChR2 stimulation should regulate intrinsic persistent firing modes. Because of the slow activation and recovery kinetics of ERG (*Wang et al., 1997a*, *Hardman and Forsythe, 2009*, *Cui and Strowbridge, 2018*), neocortical pyramidal cells remain hyperexcitable following many types of depolarizing stimuli when

mAChRs are activated. Using bath application of muscarinic receptor agonists, we showed (*Cui and Strowbridge, 2018*) that this post-stimulus hyperexcitabilty reflects a transient reduction in the leak K$^+$ current normally mediated by ERG, which can trigger persistent firing following step stimuli when neurons are maintained near their firing threshold. We believe this is the first report of ChR2-stimulated ACh release triggering persistent firing when paired with depolarizing step stimuli though this phenomenon has been widely reported following bath application of cholinergic agents in neocortical (*Krnjević et al., 1971*, *Rahman and Berger, 2011*), hippocampal (*Jochems and Yoshida, 2013*, *Knauer et al., 2013*) and entorhinal (*Klink and Alonso, 1997*, *Egorov et al., 2002*) neurons. The sensitivity of step-evoked persistent activity in our ChR2 experiments to ERG antagonists (and insensitivity to SK and M current blockers) suggests a similar underlying mechanism–transient attenuation of ERG. While few studies assayed the effects of pairing ChR2-driven ACh release with step stimuli, Baker et al. (*Baker et al., 2018*) performed this experiment and failed to find persistent firing. This difference with our results may reflect a requirement for ACh release throughout the discharge, as discussed above, as well as a potentially higher density of AAV-infected neurons with non-floxed ChR2 transcripts (versus injecting a floxed ChR2 virus in a ChAT-cre mouse, as employed in *Baker et al., 2018* and *Joshi et al., 2016*). In addition, the wild type rodents used in our study do not have the enhanced spontaneous ACh release often associated with AAV-ChR2 infected ChAT-cre mouse lines (*Kolisnyk et al., 2013*, *Chen et al., 2018*, *Hedrick et al., 2016*).

Several potential mechanisms could explain why ChR2-stimulated release of endogenous ACh selectively attenuated only one of at least three mAChR-modulated currents present in L5 pyramidal cells. First, brief pulses of ACh may trigger different signaling cascades in mAChRs than prolonged application of non-physiological receptor agonists. Previous work has identified two distinct binding sites on mAChRs (*Wamsley et al., 1997*, *Birdsall et al., 1978*) and determined that the commonly used artificial cholinergic receptor agonist CCh preferentially binds to the high affinity orthosteric site (*Larocca et al., 1987*) while endogenous ACh preferentially binds to the lower affinity allosteric site (*Kellar et al., 1985*). Ligand binding at allosteric sites appears to bias the resulting signal cascades towards predominately Gq-based versus arrestin-based mechanisms. Binding of CCh at the orthosteric site can efficiently recruit arrestin 3 (*Davis et al., 2010*), which inhibits Gq-coupled signaling transduction. It is possible, therefore, that brief pulses of endogenous ACh trigger ERG modulation via the G protein pathway while prolonged perfusion of artificial agonists like CCh modulate additional currents by recruiting the arrestin pathway. Recently, Roth and colleagues (*Wacker et al., 2017*) demonstrated a similar pattern of differential activation of signaling pathways originating at 5-HT2 receptors. In addition, the potential targets of mAChR modulation we examined are likely located within different compartments. ERG is expressed at high levels on dendrites (*Hardman and Forsythe, 2009*, *Westenbroek, 2009*) while M (KCNQ) channels are predominantly expressed on the cell body and the axons of the neocortical neurons and SK channel on dendritic spines (*Trimmer, 2015*, *Child and Benarroch, 2014*). These spatial differences may enable different subsets of cholinergic axons to selectively regulate different K$^+$ currents.

## Functional significance

By creating virtual ensembles from our ChR2 experiments, we showed that population SNR was enhanced by ACh because the SFA modulation preferentially increased the signal component while leaving the noise component unchanged. ChR2-driven ACh release also decreased pairwise signal correlations within virtual networks, an effect that we observed preferentially in response to stimuli lasting >650 ms. Both SNR and signal correlation effects appear to derive from a reduction in the ERG-mediated late phase of SFA as they were reproduced in computational models that varied the density of an ERG-like linearly increasing K$^+$ current. Neither SNR nor signal correlation effect reflects a trivial result expected from any modulation that enhances excitability. Indeed, reducing a temporally uniform K$^+$ current would not be expected to change network signal correlation.

Both SNR and signal correlation effect recapitulate commonly reported network signatures of cholinergic stimulation (*Metherate et al., 1992*, *Kalmbach et al., 2012*, *Pinto et al., 2013*, *Goard and Dan, 2009*, *Minces et al., 2017*) and attention (*Pfurtscheller et al., 1999*, *Fries et al., 2001*) in vivo. The similarity between these in vivo results and our experimental results raises the possibility that changes in intrinsic properties such as SFA could contribute to the statistical properties that distinguish different network states associated with behavioral conditions. While the potential intrinsic basis of network statistical properties has been explored in previous computational

studies (*Wang et al., 2014a*, *Crook et al., 1998*, *Ermentrout et al., 2001*), few studies have attempted to infer network statistical properties based on how experimentally-recorded intrinsic physiological properties varied with cholinergic stimulation. Instead, many computational and theoretical studies postulate synaptic and circuit modulation as potential mechanisms to reduce signal correlations (e.g., through STDP, *Min et al., 2018*, or altering excitatory/inhibitory synaptic balance, *Litwin-Kumar et al., 2011*). Computational results, however, indicate that simply reducing SFA is sufficient to decrease network signal correlation (*Wang et al., 2014b*).

Variation in SFA likely represents a biologically useful continuum that is controlled by cholinergic and other modulatory inputs. Strong SFA promotes similar discharge kinetics within networks and likely functions to enhanced decoding ability (*Cortes et al., 2012*, *Farkhooi et al., 2011*). Conversely, decreasing SFA reduces the temporal patterning of discharges, likely increasing heterogeneity among constituent neurons and enabling greater information content within networks (*Goard and Dan, 2009*). Our results indicate that different modulatory systems likely influence discharge patterning by selectively reducing individual currents involved in different phases of SFA, allowing for both simple gain control operations (e.g., by attenuating the quasi-static M current) and more complex manipulations that affect network SNR and correlation statistics when strongly time-dependent currents like ERG are modulated. In addition, selectively altering the late-developing ERG current has the additional effect of restricting the modulation to subsets of neurons–those with discharges long enough to be slowed by ERG–while leaving brief responses unaffected.

The selective modulation of ERG by endogenous ACh also may help explain the mechanisms of actions of schizophrenia treatments, many of which target cholinergic systems (*Freedman, 2014*, *Cohen, 1944*) and also modulate ERG (*Wible et al., 2005*, *Cockerill et al., 2007*). The efficacy of cholinergic agents in schizophrenia have lead many researchers to propose that dysfunction of ACh modulation underlies many aspects of this disease (*Higley and Picciotto, 2014*, *Sarter and Bruno, 1998*, *Tandon et al., 1991*). Previous genetic work implicated ERG mutations in a subset of schizophrenic patients (*Apud et al., 2012*, *Hashimoto et al., 2013*, *Atalar et al., 2010*, *Huffaker et al., 2009*). While the location of most frequent schizophrenia-associated ERG mutation is located on a non-coding (intron) site, previous work (*Huffaker et al., 2009*) demonstrated that this mutation promoted transcription from an alternative transcription start site. The isoforms resulting from this intron mutation were modified (affecting the ERG1 subunit), leading to altered gating properties and reduced ERG currents when assayed in heterologous expression systems (*Huffaker et al., 2009*, *Heide et al., 2012*). Our present results raise the possibility that some aspects of schizophrenia, especially positive symptoms such as psychosis, may relate to altered cortical excitability produced by abnormal ERG currents and suggest novel therapeutic approaches based on ERG-modulating agents (*Wible et al., 2005*).

## Materials and methods

### Slice preparation

Sprague Dawley rats of either sex were used in all experiments. In the experiments using ChR2, rats were injected with the viral vector at postnatal day 21–25 (P21-P25), and was recorded 2–4 weeks later (P35-P50). Rats aged P14-P21 were used in all other experiments not involving ChR2 optogenetics. All experiments were performed under guidelines approved by the Case Western Reserve University Animal Care and Use Committee.

### Stereotaxic injection

Sprague-Dawley rats of either sex (P21 ~ P25, 60 ~ 80 g) were deeply anesthetized with 5% xylazine/ 10% ketamine (10 mg/kg and 100 mg/kg, respectively). Under aseptic conditions, the skull was exposed and small holes with ~ 1 mm in diameter were drilled bilaterally. Stereotaxic coordinates of NB are (in mm, relative to the bregma): AP −1.4, ML 2.5 ~ 3.5, DV 6.0 (below the pial surface) (c.f. *Metherate et al., 1992*, *Paxinos and Watson, 2006*, and *Khazipov et al., 2015*). ChR2 was expressed virally in NB by microinjection of AAV1-CAG-ChR2-GFP (obtained from the UNC Vector Core). 1000 nl was injected over 10 min bilaterally using Nanoject II (Drummond Scientific Company). Injection coordinates were verified by injecting 1000 nl DiI (1 mM in DMSO) bilaterally and aligned to the stereotaxic atlas (*Paxinos and Watson, 2006*) at the corresponding coronal section

(*Figure 1—figure supplement 3*). After the surgery, rats received buprenorphine (0.03 mg/kg) twice a day for 72 hr for analgesia. We waited between 14 days and 28 days (P35 ~ P50) before doing the subsequent recordings.

## Intracellular recordings in acute brain slices

Sprague-Dawley rats were anesthetized with ketamine and decapitated. The brain was then dissected and transferred into ice-cold (~0°C) artificial CSF (ACSF) composed of the following (in mM): 124 NaCl, 2.54 KCl, 1.23 NaH$_2$PO$_4$, 6.2 MgSO$_4$, 26 NaHCO$_3$, 10 dextrose, and 1 CaCl$_2$, equilibrated with 95% O$_2$/5% CO$_2$. Horizontal slices (300 $\mu$m thick) were prepared from temporal association cortex (TeA; at the same dorsal–ventral level as the ventral hippocampus) using a Leica VT1200 Vibratome as previously described (*Cui and Strowbridge, 2018*). Slices were incubated at 30 °C for ~30 min and then maintained at room temperature (~25°C) until use.

Intracellular recordings were performed in a submerged chamber maintained at 30°C and perfused continuously (~2 ml/min) with ACSF containing the following (in mM): 124 NaCl, 3 KCl, 1.23 NaH$_2$PO$_4$, 1.2 MgSO$_4$, 26 NaHCO$_3$, 10 dextrose, 2.5 CaCl$_2$, equilibrated with 95% O$_2$/5% CO$_2$. Whole-cell recordings were made using an Axopatch 1D Amplifier (Molecular Devices/Molecular Dynamics). Patch-clamp recording electrodes with resistances of 3–8 MΩ were pulled from 1.5 mm outer diameter thin wall borosilicate glass tubing (WPI), using a micropipette puller (P-97, Sutter Instruments). The pipettes contained the following (in mM): 140 K-methylsulfate (MP Biomedicals), 4 NaCl, 10 HEPES, 0.2 EGTA, 4 MgATP, 0.3 Na$_3$GTP, and 10 phosphocreatine. In some experiments, 10 mM bis(2-aminophenoxy)ethane-N,N,N',N'-tetra-acetic acid (BAPTA) was substituted for EGTA in the internal solution to enhance Ca$^{2+}$ buffering. Individual neurons were visualized under infrared differential interference contrast (IR-DIC) video microscopy (Axioskop FS1, Zeiss). Recordings were low-pass filtered at 5 kHz (FLA-01, Cygnus Technology) and acquired at 10 kHz using a simultaneously sampling 16-bit data acquisition system (ITC-18, InstruTech) operated by custom software written in VB.NET on a Windows-based PC (*Strowbridge, 2019*, available at https://github.com/Strowbridge-Lab/ITCAcquire, copy archived at https://github.com/elifesciences-publications/ITCAcquire). Membrane potentials related to current-clamp recordings in the text and illustrations were not corrected for the liquid junction potential. Membrane potentials related to voltage clamp recordings were adjusted for the the liquid junction potential to enable comparisons with prior ERG biophysical studies. We calculated the correction factor to be + 10.3 mV based on the method outlined in *Barry (1994)*.

We recorded from L5 pyramidal cells that generated 'regular spiking' in response to 2 s duration depolarizing currents, as defined by previous neocortical studies (*Connors and Gutnick, 1990*, *Schubert et al., 2001*, *Dégenètais et al., 2002*). For 61 cells from young rats (P14-P25), the average input resistance in current clamp recordings was 82.4 ± 4.45 MOhm; the average resting membrane potential was −69.0 ± 0.95 mV. For 64 cells from adult rats with ChR2 expression, the average input resistance was 70.9 ± 3.6 MOhm; the average resting membrane potential was −66.2 ± 0.41 mV. Recordings from neurons with resting membrane potential more depolarized than −60 mV or Rin < 40 MOhm were excluded.

In voltage clamp experiments, 5 mM QX314 was included in the internal solution to block Na$^+$ currents since extracellular TTX would limit ChR2-stimulated ACh release. We used a modified ACSF with higher K$^+$ concentration (10 mM, replacing equimolar Na$^+$ as in *Sturm et al., 2005*) and a set of pharmacological ion channel blockers including 1 mM TEA, 100 $\mu$M 4-AP, 10 $\mu$M XE991 (to block I-M), 10 $\mu$M ZD7288 (to block I-H), 10 $\mu$M AFDX-116 (to block m2 subclass muscarinic receptors). Series resistance was compensated by > 80% (using 10 $\mu$s lag) in voltage-clamp experiments. We monitored access resistance continuously through each experiments by analyzing transient responses to weak hyperpolarizing steps. We excluded experiments if the access resistance increased by > 1 MOhm during the experiment. The average access resistance in the voltage-clamp experiments presented in this study was 4.57 ± 0.42 MOhm. Membrane potentials presented in the illustrations of voltage-clamp recordings were corrected for the liquid junction potential to enable comparisons with prior studies. Exponential fits of ChR2-sensitive currents (e.g., *Figure 4B*) were computed from the final 3 s of the response and then extrapolated to estimate current onset (the 0 crossing time). The magnitude of the light-sensitive outward current (*Figure 4F*) was computed from the mean of final 100 ms of the 4 s depolarizing step response. The magnitude of the light-sensitive tail current (*Figure 4G*) was computed by subtracting the mean current during the final 100 ms of

the 2 s hyperpolarizing step to −100 mV from maximal tail current evident immediately after stepping to −100 mV (measured between 25–50 ms after step onset to minimize the contribution from capacitive transients).

Salts and other routine chemicals were obtained from Sigma. Except where noted, receptor and ion channel blockers were obtained from Tocris. ErgToxin1 (STE-450) was obtained from Alomone Labs and dissolved directly in ACSF (at 50 nM final concentration).

## Optogenetic stimulation

473 nm LED light (custom built) was used to stimulate axon fibers projected from NB in neocortical slices through the epifluorescence pathway. LED illumination focused directly on the recorded neuron through 63x Zeiss water objective (NA 0.9) with no restriction through the field diaphragm. Brief (2 ms) light pulses were controlled by the TTL output of the ITC-18 to stimulate ChR2. Throughout the illustrations, orange traces denote experiments that included ChR2 stimulation while blue traces and bars represent experiments using K$^+$ channel blockers and green traces and bars represent experiments using bath application of CCh.

## Computational modeling

Hodgkin-Huxley (HH) model of neurons was built based on *Rothman and Manis (2003)*, and was implemented using Brian2 (*Stimberg et al., 2013*). State and kinetic parameters of our model are identical to that of *Rothman and Manis (2003)*. Conductance parameters are modified as the following (in *nS*): $\bar{g}_{Na} = 40000$, $\bar{g}_{K_{HT}} = 13000$, $\bar{g}_{K_{LT}} = 0$, $\bar{g}_{K_A} = 3325$, $\bar{g}_h = 1$, $\bar{g}_{no} = 0$ (*h*-current of octopus cell), $\bar{g}_l = 10$, such that we can produce similar number of spikes to our intracellular recordings given a step current of similar amplitude used in our experiments.

To introduce variability in spike timing, noisy conductance term $\epsilon$ was specified by a Langevin equation (*Schmidt et al., 2015*, *Moreno et al., 2002*, *Ratte et al., 2015*), denoted by the stochastic differential equation $\frac{d\epsilon}{dt} = -\frac{\epsilon}{\tau_\epsilon} + \frac{\xi}{\sqrt{\tau_\epsilon}}$, where $\xi \sim \mathcal{N}(0, 1)$ (Gaussian of mean 0 and standard deviation of 1). Here, $\tau_\epsilon$ is a hyperparameter and is set to 30 ms unless otherwise stated. Following the HH model form, we scaled the noise term with the factor $\sigma_\epsilon(E_K - V)/(E_K - E_l)$, where $\sigma_\epsilon$ is a hyperparameter (set to 1.05 unless stated otherwise), $E_K$ is the reversal potential of K$^+$ currents, $V$ is the membrane potential, $E_l$ is the resting membrane potential, or the reversal potential of the leak currents.

Artificial K$^+$ currents were added onto the HH model with $I_{manK} = \bar{g}_{manK} \cdot g_{manK}(t) \cdot (E_K - V)$, where $\bar{g}_{manK}$ is the maximum conductance of the artificial K$^+$ current, and $g_{manK}(t)$ is the specific time series of artificial K$^+$ current conductance with maximum amplitude of 1. 'No K' denotes no artificial K$^+$ current was added to the HH model. 'Linear K' denotes linearly increasing artificial K$^+$ current, with slope $4\mu S$ per second (for a 2 s stimulus, the max conductance is $8\mu S$). 'Continuous K' denotes a 8 $\mu$S step increase of artificial K$^+$ current during the entire period of the stimulus.

## Experimental design and statistical analysis

We used the following procedure to determine the time when the number of spikes in the experimental conditions exceeds that of control conditions by at least 1 ($\Delta 1AP$ time). We divided the 2 s window during the depolarizing current step into 40 non-overlapping bins, each of 50 ms. We then count the cumulative number of spikes since the beginning in each bin. The center of the bin (i.e. 25, 75, 125, etc.) was used as the time of the bin. An example of this process is illustrated in *Figure 2A*. Bifurcation time was subsequently determined as the first time bin in which the average number of spikes in the experiment condition exceeds the number of spikes in the control condition by 1. To determine the rate of change of spike frequency adaptation modulated by ACh and different K$^+$ currents, we calculated the metric that we termed 'Slope' (*Figure 6*, using the same set of data as in *Figure 5C–D*). Difference time series were calculated by subtracting control time series from experimental time series. To determine the rate of change of SFA over time, we determined the slope using a moving 300 ms (six bins) window (*Figure 6A–C*). We also quantified specifically the slopes at the beginning and the end of the step (*Figure 6D*). To calculate the early rate of change of number of spikes (early-slope), the first two data points of the difference time series (i.e. first 100 ms) were used and the slope was obtained from the fitted straight line. To calculate the late-slope, the last six data points of the difference time series (last 300 ms) were used. To calculate adaptation ratio, we take the ratio between the average of the last 2 ISIs and the average of the first 2 ISIs.

We estimated dynamic changes in input resistance (*Figure 8D*) using the same two-step correction method we employed in *Cui and Strowbridge (2018)*. Summary Rin plots show the mean ± SEM across multiple experiments of these corrected Δ Rin estimates. Except where noted in figure legends, data were expressed as the mean ± SEM. Significance level with p<0.05 was used. Multiple comparisons using t tests were Bonferroni corrected.

Signal-to-noise ratio (SNR) was calculated by the inverse of the coefficient of variance (i.e. mean over standard deviation of spike counts across trials). Methods used to estimate signal correlation were based on approach used by *Minces et al. (2017)* and *Goard and Dan (2009)* to analyze extracellularly recorded spike trains in vivo. Briefly, for each trial, we divided the spike train with 50 ms non-overlapping bins and counted the number of spikes within the bin. For the ChR2 data set and simulation data where we injected 2000 ms of step current, total 40 bins were returned. Average binned spike count is then calculated by averaging across all the trials available. Therefore, for each cell, we have one time series of binned spike counts. We then calculate the signal correlation between each pair of neurons using these time series bins. An illustration of this calculation process is shown in *Figure 10B–C* (and see *Minces et al., 2017*).

## Acknowledgments

We thank Diana Kunze, Todd Pressler, Joel Zylberberg and Chris Ford for helpful conversations related to this project. This work supported by NIH grant R01-DC004285 to BWS.

## Additional information

### Funding

| Funder | Grant reference number | Author |
| --- | --- | --- |
| National Institute on Deafness and Other Communication Disorders | R01-DC04285 | Ben W Strowbridge |

The funders had no role in study design, data collection and interpretation, or the decision to submit the work for publication.

### Author contributions

Edward D Cui, Conceptualization, Data curation, Software, Formal analysis, Supervision, Funding acquisition, Validation, Investigation, Visualization, Methodology, Writing—original draft, Project administration, Writing—review and editing; Ben W Strowbridge, Conceptualization, Data curation, Software, Formal analysis, Validation, Investigation, Visualization, Methodology, Writing—original draft, Writing—review and editing

### Author ORCIDs

Ben W Strowbridge https://orcid.org/0000-0002-3761-6226

### Ethics

Animal experimentation: This study was performed in strict accordance with the recommendations in the Guide for the Care and Use of Laboratory Animals of the National Institutes of Health. All of the animals were handled according to approved institutional animal care and use committee (IACUC) protocols of Case Western Reserve University. The protocol supporting this work (2016-0020) was approved by the Committee on the Ethics of Animal Experiments of CWRU. All surgery was performed under anesthesia, and every effort was made to minimize suffering.

### Decision letter and Author response

Decision letter https://doi.org/10.7554/eLife.44954.028
Author response https://doi.org/10.7554/eLife.44954.029

## Additional files

### Supplementary files

• Transparent reporting form
DOI: https://doi.org/10.7554/eLife.44954.026

### Data availability

All data generated or analysed during this study are included in the manuscript and supporting files. Custom analysis routines and scripts are available https://github.com/StrowbridgeLab/ITCAcquire (copy archived at https://github.com/elifesciences-publications/ITCAcquire).

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
