## [Decision Letter]

Thank you for submitting your article "Selective attenuation of ERG by endogenous acetylcholine reduces spike-frequency adaptation and network correlation" for consideration by *eLife*. Your article has been reviewed by Richard Aldrich as the Senior Editor, a Reviewing Editor, and three reviewers. The following individuals involved in review of your submission have agreed to reveal their identity: Lorna Role (Reviewer #1).

The reviewers have discussed the reviews with one another and the Reviewing Editor has drafted this decision to help you prepare a revised submission.

The reviewers find that the manuscript describes a timely and interesting set of results regarding the mechanisms by which synaptically released ACh increases the excitability of L5 pyramidal neurons. The authors show that optogenetic stimulation of ACh fibers increases firing frequency, and using pharmacological approaches they argue that the target of ACh is hERG channels. There have been relatively few electrophysiological staudies of the role of hERG in the CNS despite its widespread distribution, and this paper suggests that these channels may be an important target. The authors speculate that acetylcholine and depolarization-dependent attenuation of ERG channels may account for acetylcholine's hypothesized effects on persistent activity, working memory, and signal-to-noise ratio.

Essential revisions:

The reviewers raise a number of concerns that must be adequately addressed before the paper can be accepted. Some of the required revisions will likely require further experimentation.

1) The identification of hERG as the key target of ACh is not air-tight. Experiments with bath applied ACh or agonists have shown previously that muscarinic stimulation activates calcium-dependent non-specific cation conductances as well as blocking M-channels. Here the hERG current in these cells is not isolated in voltage-clamp recordings that could support the conclusion that hERG is present in these cells and acting late to modulate spike-frequency adaptation (e.g., Carver and Shapiro, 2018). It will be critical for the authors to demonstrate that synaptically released ACh specifically blocks the hErg mediated current in voltage-clamp, providing direct evidence for (1) the modulation of the hErg current by ACh and (2) the time course of ACh-induced block of Erg currents, which currently the authors only indirectly inferred.

2) Terfenedine may not be the best choice as an identifier of hERG currents as its primary target is H1 receptors. The use of the more specific compound, E4031, is a useful addition, but some of these experiments have an unacceptably low n (e.g., 1). The n's must be increased for this essential experiment.

3) In some figures, data from experiments with reasonable n values are presented only as bar graphs. The preferred presentation is to show a full scatter plot of all of the data because it's useful for readers to see individual data points to understand variability more clearly than standard error alone.

4) The modeling adds only modestly to the present work. Its contribution would be greater if the authors could explain: (a) why a one-compartment model of ventral cochlear nucleus neurons helps us understand layer 5 pyramidal neurons with elaborate dendrites; (b) how a model that does not contain BK, SK, or M channels deepens our understanding of SFA; (c) in what way putting in by hand a time-dependent K^+^ conductance ("linear K") to get a time-dependent SFA effect proves anything; and (d) why adding stochastic noise directly to the membrane potential is better than adding stochastic noise to the *conductances* - since real background fluctuations in vivo are mediated by conductance fluctuations.

The modeling should be strengthened considerably or simply eliminated.

---

## [Author Response]

Essential revisions:The reviewers raise a number of concerns that must be adequately addressed before the paper can be accepted. Some of the required revisions will likely require further experimentation.1) The identification of hERG as the key target of ACh is not air-tight. Experiments with bath applied ACh or agonists have shown previously that muscarinic stimulation activates calcium-dependent non-specific cation conductances as well as blocking M-channels. Here the hERG current in these cells is not isolated in voltage-clamp recordings that could support the conclusion that hERG is present in these cells and acting late to modulate spike-frequency adaptation (e.g., Carver and Shapiro, 2018). It will be critical for the authors to demonstrate that synaptically released ACh specifically blocks the hErg mediated current in voltage-clamp, providing direct evidence for (1) the modulation of the hErg current by ACh and (2) the time course of ACh-induced block of Erg currents, which currently the authors only indirectly inferred.

We conducted three additional sets of experiments to address this point. First, we tested two additional ERG blockers (E-4031 and ErgToxin) using the same current-clamp protocol presented in the R0 version. Both E-4031 and ErgToxin1 blocked persistent firing, as we reported previously for terfenadine. While terfenadine has actions on other targets besides ERG, ErgToxin1 and E-4031 are highly specific for ERG channels. These current-clamp experiments also set up the new voltage-clamp studies employing E-4031 that the reviewers requested. We also used this battery of three ERG blockers in our previous JN paper on ERG modulation of persistent firing by bath carbachol and used terfenadine to determine ERG current kinetics and modulation by ACh under voltage-clamp conditions.

Second, we performed voltage-clamp experiments to define the properties of K current modulated by ChR2 stimulation (new Figure 4). These recordings demonstrated that ACh release reduced an outward current with very slow kinetics that appeared to initiate after ~0.5 seconds of depolarization. Both the late onset and slow kinetics of the ACh-sensitive current fit well with properties we inferred from our analysis of how spike-frequency adaptation changed with ChR2 stimulated ACh release and, in separate experiments, when ERG current was attenuated by terfenadine. We also show that the ERG blocker suggested by the reviewers, E-4031, greatly reduced the ACh-sensitive slow outward current, providing additional evidence supporting our hypothesis that ERG is a target of ACh modulation.

Finally, we conducted additional current-clamp experiments testing whether blockade of other K currents could interfere with persistent firing. Consistent with our previous results, we now report that SK attenuation (using NS8593, now N=3) could not block persistent firing triggered by coincident ACh release and intracellular depolarization. Together, these studies provide additional evidence for ERG as a target of ACh modulation and also define time course of the underlying outward current, as requested by the reviewers.

2) Terfenedine may not be the best choice as an identifier of hERG currents as its primary target is H1 receptors. The use of the more specific compound, E4031, is a useful addition, but some of these experiments have an unacceptably low n (e.g., 1). The n's must be increased for this essential experiment.

As outlined in response #1 above, we conducted additional experiments using E-4031, the antagonist suggested by the reviewers, as well as the ERG-specific peptide blocker ErgToxin1 that we used in previous paper on ERG in JN. We also clarified in the R1 text that we had previously tested fexofandine, another H1 receptor blocker that is structurally similar to terfenadine but does not affect ERG channels. We reported in our JN paper last year that it did not affect persistent activity even when used at 3x times the concentration used in our terfenadine experiments. We also increased the Ns of the SK blocker experiment that was flagged to 3 (all resulted in negative findings with no blockade of persistent firing.)

3) In some figures, data from experiments with reasonable n values are presented only as bar graphs. The preferred presentation is to show a full scatter plot of all of the data because it's useful for readers to see individual data points to understand variability more clearly than standard error alone.

We have reformatted most of the figure panels that were bar graphs to include raw data points. In most of the light/no-light comparisons we have also included connecting lines to allow the reader to see corresponding data points in each experiment. We also retained some bar graphs without including individual data points when the means represented a very large number of trials or when the goal of the illustration was to illustrate how similar multiple mean values were rather than to make a claim about statistically significant differences between means.

4) The modeling adds only modestly to the present work. Its contribution would be greater if the authors could explain: (a) why a one-compartment model of ventral cochlear nucleus neurons helps us understand layer 5 pyramidal neurons with elaborate dendrites; (b) how a model that does not contain BK, SK, or M channels deepens our understanding of SFA; (c) in what way putting in by hand a time-dependent K^+^ conductance ("linear K") to get a time-dependent SFA effect proves anything; and (d) why adding stochastic noise directly to the membrane potential is better than adding stochastic noise to the *conductances* – since real background fluctuations in vivo are mediated by conductance fluctuations.The modeling should be strengthened considerably or simply eliminated.

We included the modelings results in the original manuscript for two different reasons. First, to provide easily understood reference data sets to illustrate the analysis methods we developed to define modulation in spike-frequency adaptation (the cumulative spike count panels and the slope methods for estimated the kinetics of the ACh-modulated current). We prefer to retain these panels in the main illustrations as they provide helpful illustration of the analysis work flow that would be difficult to replace using real biological data. The second application of the model was to demonstrate that we could replicate the network ensemble changes that we find occur with ACh release. We agree with the reviewers that the unusual nature of the model implementation we used (i.e., K currents with pre-defined kinetics rather than traditional HH kinetics) limits the additional impact of these simulations without including more work supporting this approach. We therefore moved the network simulation results that were in Figure 8G,H to Figure 10—figure supplement 1 and deemphasized the conclusions from modeling work.